# Reactive oxygen species in the rhizosphere orchestrate the recruitment of beneficial bacteria

Xijie Guo[1,6], Hengyi Dai [2,3,4,6], Zhiyi Jia[1], Ying Peng[1], Luotian Lu[2,3,4], Yaxing Su[1], Jianwei Li [1], Qinghong Li[1], Zeming Huang[1], Yucheng Wang[1], Fan Qi[1], Dayong Li[1], Xiaofei Lv [5], Yan Liang [1✉] & Bin Ma [2,3,4✉]

## Abstract

Respiratory burst oxidase homolog D (RBOHD)-dependent reactive oxygen species (ROS) in *Arabidopsis* are well known to suppress pathogen colonization, but their influence on beneficial microbes remains unclear. Here, we found that the beneficial rhizobacterium *Pseudomonas anguilliseptica* was significantly less enriched in the rhizosphere of *rbohD* mutants than in that of wild-type plants. Conversely, elevated rhizosphere ROS levels, either triggered by pretreatment with pathogenic *Dickeya solani* bacteria or caused by mutations in ROS scavenging genes (e.g., in *apx1* and *cat2* mutants), promoted the rhizosphere recruitment of *P. anguilliseptica*. This promoting effect was abolished by catalase treatment. In situ microfluidic chemotaxis assays further revealed that *P. anguilliseptica* exhibits a chemotactic response to low concentrations of hydrogen peroxide ($\leq 500$ nM), accompanied by upregulated expression of chemotaxis- and motility-related genes. Notably, inoculation of *P. anguilliseptica* effectively suppressed *D. solani*-induced disease symptoms, and this protective effect was attenuated by catalase treatment. Collectively, these findings reveal a previously unrecognized role of ROS in recruitment beneficial microbiota to enhance plant growth and suppress disease symptoms.

**Keywords** Plant Immunity; Microbiome; Rhizosphere ROS; Microfluidic Chip
**Subject Categories** Microbiology, Virology & Host Pathogen Interaction; Plant Biology

## Introduction

Reactive oxygen species (ROS) are highly reactive oxygen-containing compounds, including singlet oxygen, hydroxyl radicals, superoxide anions, and hydrogen peroxide ($H_2O_2$) (Mittler et al, 2022). Among these, $H_2O_2$ exhibits moderate stability, enabling transmembrane diffusion or facilitated transport via aquaporins (Rodrigues et al, 2017; Tian et al, 2016). ROS production is a critical immune response in plants that defends them against pathogens (Jones et al, 2024; Wang et al, 2024). In *Arabidopsis*, ROS are predominantly generated by respiratory burst oxidase homolog D (RBOHD), a nicotinamide adenine dinucleotide phosphate (NADPH) oxidase (Torres et al, 2002). RBOHD consists of six transmembrane domains, with a C-terminal NADPH-binding domain located in the cytoplasm (Kawahara et al, 2007; Segal, 2008). Upon activation, RBOHD transfers electrons from cytosolic NADPH across the plasma membrane to the apoplast, where oxygen is reduced to superoxide and subsequently converted to $H_2O_2$ by superoxide dismutase (Wu et al, 2022; Wu et al, 2023). ROS function as both antimicrobial agents and secondary messengers, coordinating downstream immune processes such as cell wall fortification and systemic acquired resistance (Cao et al, 2024; Fujita et al, 2020; Liao et al, 2025).

The rhizosphere microbiome is often referred to as "second genome" due to the indispensable role played by microbial communities in promoting plant growth and health (Berendsen et al, 2012). During pathogen challenge, plants activate immune responses that not only limit microbial invasion but also alter exudate composition, including polysaccharides, proteins, amino acids, organic acids, phytohormones, and phenolic compounds, which in turn facilitate the recruitment of beneficial microbes (Trivedi et al, 2020; Afridi et al, 2024). Such beneficial microorganisms provide multilayered protection against pathogens through ecological niche competition, growth modulation, and immune response priming (Van Elsas et al, 2012; Liu et al, 2025; Zamioudis and Pieterse, 2012). Previous studies have demonstrated altered microbiome composition in *rbohD* mutant leaves compared to wild-type plants (Pfeilmeier et al, 2021). However, whether ROS directly modulates microbiome assembly or indirectly influences it through ROS-mediated changes in plant development and exudation patterns remains unclear.

In this study, we investigate the role of rhizosphere ROS in plant–microbe interactions. Comparative analysis of rhizobacterial communities revealed significant compositional differences

[1]Zhejiang Provincial Key Laboratory of Agricultural Microbiomics, Key Laboratory for Agricultural Microbiome of the Ministry of Agriculture and Rural Affairs, Institute of Biotechnology, Zhejiang University, Hangzhou 310058, China. [2]State Key Laboratory of Soil Pollution Control and Safety, Zhejiang University, Hangzhou 310058, China. [3]ZJU-Hangzhou Global Scientific and Technological Innovation Center, Zhejiang University, Hangzhou 311215, China. [4]Zhejiang Provincial Key Laboratory of Agricultural, Resources and Environment, College of Environmental and Resource Sciences, Zhejiang University, Hangzhou 310058, China. [5]Department of Environmental Engineering, China Jiliang University, Hangzhou 310018, China. [6]These authors contributed equally: Xijie Guo, Hengyi Dai.✉E-mail: yanliang@zju.edu.cn; bma@zju.edu.cn

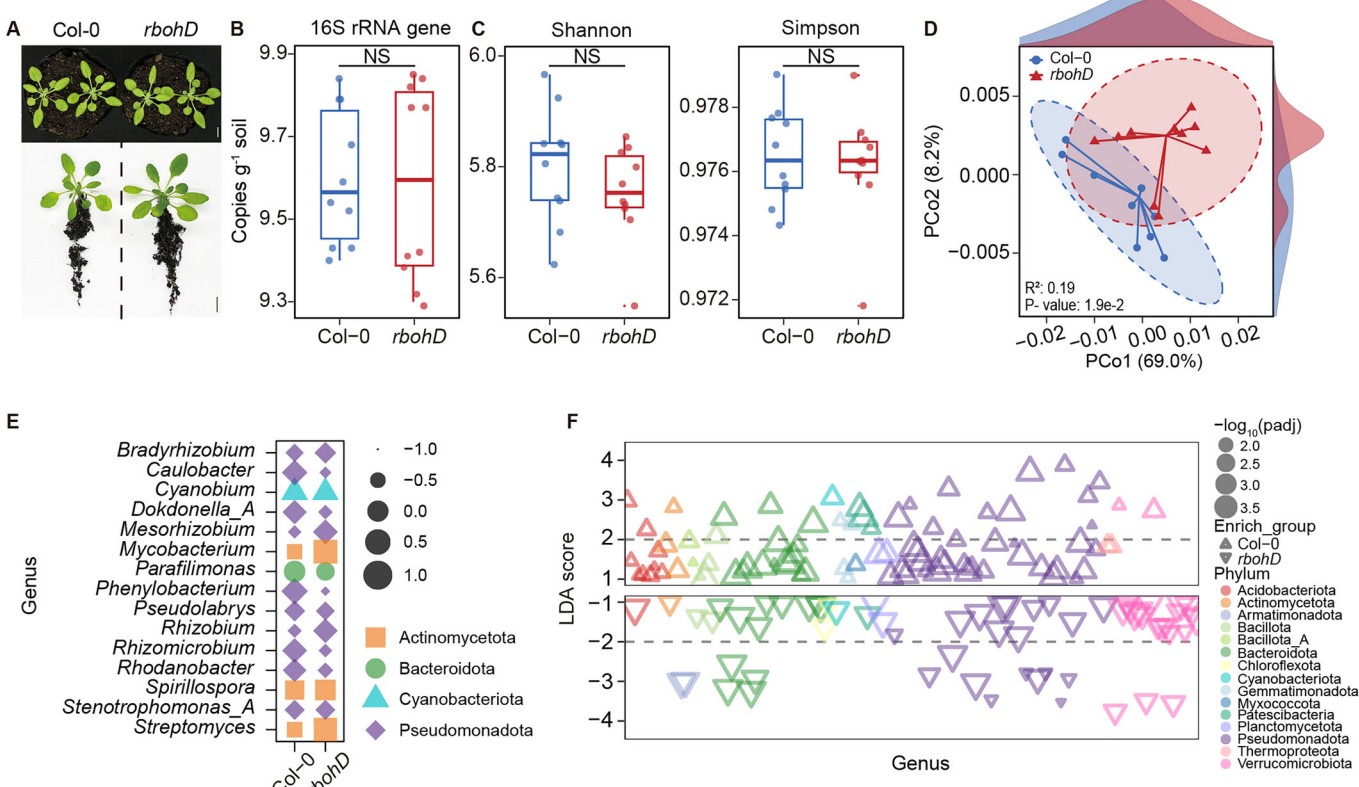

**Figure 1. RBOHD-derived ROS play a role in the assembly of the rhizosphere microbiome.**

(A) Representative images of three-week-old *Arabidopsis* wild-type (Col-0) and *rbohD* plants. Scale bar, 1 cm. (B) Bacterial 16S rRNA gene copies. Bacterial DNA was extracted from the rhizosphere soil, and 16S rRNA gene copy numbers were determined using quantitative PCR. (C) Alpha diversity based on Shannon and Simpson indices. The 16S sequences were denoised, and zOTUs were generated using UNOISE3. Box plots in (B, C) display the median, upper, and lower quartiles, with whiskers extending up to 1.5 times the interquartile range ($n = 10$ biological replicates, Student's *t*-test, NS no significant difference). (D) Beta diversity was analyzed using Weighted UniFrac dissimilarity analysis ($P = 1.9$e-2, PERMANOVA, Adonis). Ellipses represent 100% of the data ($n = 10$ biological replicates). (E) Relative abundance of bacteria at the genus level, scaled by Z-score. While circle size indicates the magnitude of the Z-score, color denotes the corresponding bacterial genus level. For each genus, Z-scores were calculated across samples. A negative Z-score indicates that the abundance in a given sample is lower than the mean abundance of that genus across all samples, whereas a positive Z-score indicates a higher-than-mean abundance. (F) The differential taxa between Col-0 and *rbohD* at the genus level based on the LDA effect size score. Each triangle represents a unique zOTU, with its corresponding phylum depicted in various colors, and the size of each triangle corresponds to the $-\log_{10}(P \text{ value})$ ($P < 0.01$, Wilcoxon). Source data are available online for this figure.

between wild-type and *rbohD* mutant plants, with the mutant rhizosphere showing altered colonization of pathogens and beneficial rhizobacteria (e.g., *Dickeya solani* increased and *Pseudomonas anguilliseptica* decreased). We further examined the chemotactic response of *P. anguilliseptica* to ROS and its protective effects against *D. solani*-induced disease symptoms. Together, these findings demonstrate that rhizosphere ROS recruit beneficial microbiota members, thereby contributing to disease suppression.

## Results

### The *rbohD* mutants exhibit altered rhizosphere microbiome assembly

To identify microbes influenced by RBOHD-derived ROS in the rhizosphere, we collected rhizosphere soil from three-week-old roots of wild-type (Col-0) and *rbohD* mutant plants (Fig. 1A). Quantitative PCR quantification of 16S ribosomal RNA (rRNA) gene copies showed

no significant difference in bacterial abundance between wild-type and *rbohD* mutants (Fig. 1B). Microbial community composition was further analyzed using 16S rRNA gene sequencing. We found that α-diversity exhibited no significant variation between the wild-type plants and *rbohD* mutants, as measured by the Shannon and Simpson indices (Fig. 1C). However, β-diversity analysis revealed notable differences in bacterial composition (Fig. 1D). Subsequently, we identified 11,937 bacterial species by assembling and annotating metagenomic contigs (Table EV1). Among these, 347 species decreased in abundance in *rbohD* mutants, whereas 1554 species increased (Appendix Fig. S1A). Z-score normalization of microbial abundance data revealed pronounced taxonomic shifts between wild-type and *rbohD* mutant groups at both the phylum and genus levels. At the phylum level, Actinomycetota, Bacillota, Bacteroidota, Bdellovibrionota, and Verrucomicrobiota were enriched, whereas Acidobacteriota, Cyanobacteriota, Myxococcota, Planctomycetota, and Pseudomonadota were depleted in *rbohD* mutants (Appendix Fig. S1B). At the genus level, *Bradyrhizobium*, *Mesorhizobium*, *Mycobacterium*, *Rhizobium*, and *Streptomyces* were enriched, while

*Caulobacter*, *Dokdonella_A*, *Parafilimonas*, *Phenylobacterium*, *Rhizomicrobium*, and *Rhodanobacter* were depleted in *rbohD* mutants (Fig. 1E).

To further investigate these variations, we performed linear discriminant analysis (LDA) coupled with effect size analysis (LEfSe) to identify differentially enriched microbial taxa (Fig. 1F). Based on LDA scores, the ten genera were identified as genotype-specific microbial markers for Col-0 and *rbohD*, respectively (Appendix Fig. S1C). Species-level co-occurrence networks constructed from metagenomic profiles revealed markedly greater complexity in the *rbohD* rhizosphere, with 7837 nodes and 158,934 edges compared to 5386 nodes and 101,645 edges in the wild type (Appendix Fig. S1D). The *rbohD* network also exhibited longer average path lengths and greater modularity, suggesting increased interconnectivity and compartmentalization. In addition, microbial genes involved in methane, nitrogen, phosphorus, and sulfur cycling, as well as virulence factors, exhibited differential expression between wild-type and *rbohD* mutants (Appendix Fig. S2). Collectively, these results suggest that loss of RBOHD function leads to distinct compositional changes in rhizosphere microbiome assembly.

## Rhizosphere ROS contribute to the enrichment of *P. anguilliseptica*

We subsequently analyzed differential beneficial and pathogenic microbes between wild-type and *rbohD* mutants to identify ROS-regulated specific microbes involved in plant growth. Metagenomic contigs were annotated using PHI-base (https://www.phi-base.org), a pathogen-host interaction database, and the Probio database (https://bidd.group/probio/homepage.htm), which catalogs published plant-beneficial bacterial species (Winnenburg et al, 2007). Compared to the wild-type, *rbohD* mutants exhibited a significant shift in microbial abundances, with seven beneficial species decreasing and 35 pathogenic species increasing (Appendix Fig. S3).

Increased abundance of *Xanthomonas* pathogens has been reported in the *rbohD* phyllosphere (Entila et al, 2024); however, no such increase was found in the rhizosphere. Instead, we observed enrichment of *D. solani*, a major causal agent of soft rot and blackleg in several crops (Matilla et al, 2023). We isolated a strain of *D. solani* from the rhizosphere and confirmed its identity using 16S rRNA and *dnaX* gene sequencing (Chen et al, 2020). Plant inoculation indicated that *D. solani* caused *Arabidopsis* soft rot symptoms (Appendix Fig. S4A–E). To examine whether increased colonization of *D. solani* occurred in the rhizosphere of *rbohD* mutants compared to the wild-type, we created a green fluorescent protein (GFP)-labeled *D. solani* strain by introducing a GFP-expressing plasmid that had no effect on bacterial growth (Appendix Fig. S4F). Indeed, pronounced green fluorescence was observed on the root surfaces of the *rbohD* mutants (Appendix Fig. S4G). Flow cytometry analysis indicated that the *rbohD* mutants exhibited significantly higher levels of green fluorescence than the wild-type (Appendix Fig. S4G–I). These results indicate that the mutation of RBOHD results in an increase of *D. solani* in the rhizosphere.

In contrast, seven beneficial bacterial species belonging to *Paenibacillus*, *Pseudomonas*, *Sphingomonas*, and *Rahnella* were reduced in *rbohD* mutants (Appendix Fig. S3). To further explore

the potential reasons why RBOHD dysfunction caused a decline in beneficial rhizobacteria, we isolated and characterized a collection of bacterial strains from key genera such as *Bacillus*, *Clostridium*, *Pseudomonas*, *Niallia*, *Acinetobacter*, *Pantoea*, *Enterobacter*, and *Massilia*. Subsequent comparative analysis revealed that, among these isolates, only *P. anguilliseptica* was consistently identified as a member of the beneficial rhizobacterial community that exhibited significantly reduced abundance in the *rbohD* mutant (Alaa, 2018). We identified it as *P. anguilliseptica* through draft genome analysis (Fig. EV1A). Seedling and soil inoculation revealed that *P. anguilliseptica* significantly increased seedling fresh weight and rosette leaf size in wild type but not in *rbohD* mutants (Figs. 2A and EV1B, C).

To test whether the reduction of *P. anguilliseptica* in the *rbohD* rhizosphere was due to competitive exclusion by pathogens, such as *D. solani*, we conducted a co-inoculation experiment with GFP-labeled *P. anguilliseptica* (Figs. 2B and EV1D). GFP-labeled *P. anguilliseptica* abundance in the rhizosphere was quantified by flow cytometry at 2 d post-inoculation (dpi). *Ensifer sesbaniae*, a beneficial nitrogen-fixing bacterium (Wang et al, 2022) whose abundance remained unchanged between wild type and *rbohD* mutants, served as a control (Table EV1). Our findings revealed that GFP-labeled *P. anguilliseptica* abundance was significantly higher in the presence of *D. solani* than in the presence of *E. sesbaniae* (Fig. 2C). These results suggest that the reduced abundance of *P. anguilliseptica* in *rbohD* mutants is not directly attributable to competition with *D. solani*.

To investigate whether the abundance of *P. anguilliseptica* in *rbohD* mutants correlates with rhizosphere ROS levels, we employed a custom-designed microfluidic imaging device to visualize rhizosphere ROS (Fig. EV2A). *Arabidopsis* seedlings were grown directly within the soil-filled chamber of the chip. Once the roots had extended and fully embedded themselves in the surrounding soil, the bottom layer of the chamber was carefully replaced with a solid agar gel filled with 2′,7′-dichlorofluorescein (H$_2$DCFDA), a ROS sensitive fluorescent dye. After a 15-min incubation period and subsequent removal of the culture layer. Green fluorescence was subsequently observed at the root position, with gradient signals extending into the rhizosphere, indicating the diffusion of root-derived ROS into the rhizosphere environment (Fig. EV2B). Following inoculation with GFP-labeled *P. anguilliseptica*, fluorescent signals were weaker in *rbohD* mutants than in wild-type plants (Fig. 2D). In contrast, GFP-labeled *P. anguilliseptica* abundance was significantly higher in the *ascorbate peroxidase 1* (*apx1*) and *catalase 2* (*cat2*) mutants, which are defective in ROS scavenging and therefore accumulate higher ROS levels (Fig. 2E). Exogenous catalase treatment markedly suppressed GFP-labeled *P. anguilliseptica* proliferation in the rhizosphere across all genotypes (Fig. 2E), whereas the bovine serum albumin (BSA) control had no such effect. Neither catalase nor BSA affected *P. anguilliseptica* growth in vitro at the tested concentrations (Appendix Fig. S5). Together, these results suggest that rhizosphere ROS directly promote the enrichment of *P. anguilliseptica*.

## Rhizosphere ROS recruit *P. anguilliseptica* toward the roots

The aforementioned findings prompted us to hypothesize that rhizosphere ROS directly attract *P. anguilliseptica* toward the roots.

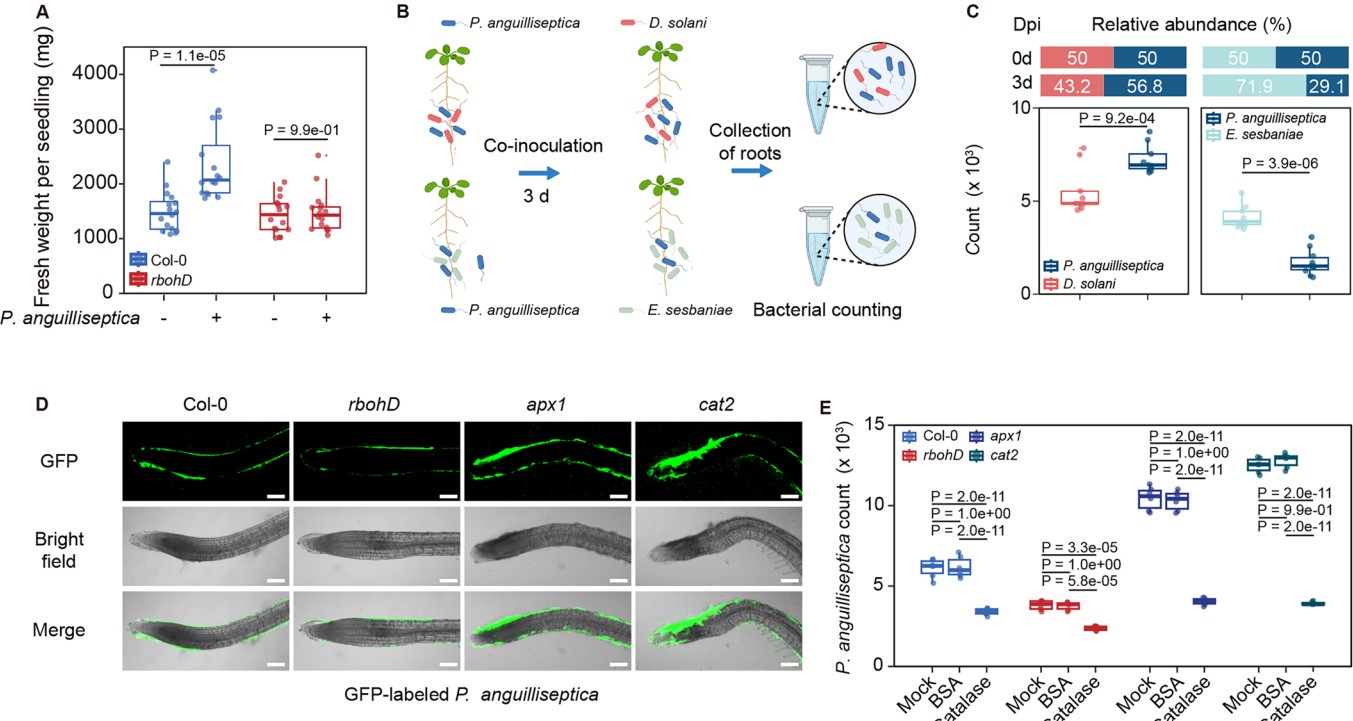

**Figure 2. RBOHD-derived ROS contribute to the enrichment of beneficial bacteria in the rhizosphere.**

(A) *Pseudomonas anguilliseptica*-induced enhancement of seedling fresh weight in Col-0, but not in *rbohD*. Box plots display the median, upper, and lower quartiles, with whiskers extending up to 1.5 times the interquartile range (*n* = 16 biological replicates). *P* values are determined (ANOVA and Tukey test). (B) Schematic diagram of the co-inoculation assay. (C) Co-inoculation assay. Ten-day-old seedlings were co-inoculated with GFP-labeled *P. anguilliseptica* and either *D. solani* or *E. sesbaniae* at equal cell densities ($OD_{600}$ = 0.001). Bacterial populations were quantified by flow cytometry 3 dpi (*n* = 9 biological replicates). *P* values are determined (ANOVA and Tukey test). (D, E) Root colonization of GFP-labeled *P. anguilliseptica* in Col-0, *rbohD*, *apx1*, and *cat2*, with or without catalase treatment. Representative images are shown in (E) and bacterial populations were quantified by flow cytometry at 3 dpi (E). Scale bar, 100 μm. Box plots display the median, upper, and lower quartiles, with whiskers extending up to 1.5 times the interquartile range (*n* = 6 biological replicates). *P* values are determined (ANOVA and Tukey test). Source data are available online for this figure.

To test this hypothesis, bacterial chemotaxis was assessed using a microfluidic chip-based in situ chemotaxis assay (Lin et al, 2024). Various concentrations of $H_2O_2$ were injected into each well through a port. Upon loading the *P. anguilliseptica* bacterial suspension, $H_2O_2$ diffused from the port, creating a chemical microplume above each well, which induced chemotaxis and caused the bacteria to swim into the well. The number of bacteria in each well was then quantified by flow cytometry (Figs. 3A and EV3A). Our results indicated that bacterial numbers in wells containing 100 nM and 500 nM $H_2O_2$ were significantly higher than in the control, whereas concentrations exceeding 1 μM repelled bacterial migration (Fig. 3B). In contrast, 100 nM $H_2O_2$ inhibited *Pantoea dispersa* chemotaxis (Fig. EV3B). These findings suggest that 100 nM $H_2O_2$ may promote the chemotactic movement of *P. anguilliseptica*. Consistent with this hypothesis, we observed that 100 nM $H_2O_2$ treatment significantly enhanced *P. anguilliseptica* motility after 2 h (Figs. 3C and EV3C), and subsequently promoted biofilm formation upon prolonged exposure (Fig. EV3D). Furthermore, we performed transcriptomic analysis on bacteria treated with 100 nM $H_2O_2$, which revealed 358 upregulated and 314 downregulated genes at significant levels (Fig. 3D). Notably, key chemotaxis-related genes (*aer2* and *cheA3*) and motility-associated genes (*flgC*, *flgE*, *flgG*, and *fliQ*) were highly upregulated after $H_2O_2$ treatment (Fig. 3E). The upregulation of

*aer2*, *cheA3*, and *flgE* were subsequently confirmed by qRT-PCR (Fig. 3F). To examine whether this upregulation also occurs during plant colonization, we analyzed bacterial gene expression by isolating *P. anguilliseptica* from wild-type and *rbohD* mutant roots. Notably, bacteria colonizing *rbohD* roots exhibited significantly lower expression of *aer2*, *cheA3*, and *flgE* than those from wild-type roots (Fig. 3G). Collectively, these findings suggest that rhizosphere ROS may serve as a chemotactic cue facilitating the recruitment and colonization of *P. anguilliseptica* in the rhizosphere.

## $H_2O_2$ treatment enhances the biosynthesis of 5-aminolevulinic acid in *P. anguilliseptica*

To elucidate the role of ROS in recruiting *P. anguilliseptica* to plant roots, we performed functional gene differential analysis of the transcriptomic data described above (Fig. 3D). A total of 146 metabolism-related genes were upregulated following $H_2O_2$ treatment, with enrichment observed in pathways associated with amino acid metabolism (*ald*, *trpA*, *trpB*, *gshA*, *gshB*, and *tyrB*), carbohydrate metabolism (*pdhA*, *glnA*, *pckA*, *gabD*, and *aceE*), energy metabolism (*cysH*, *cysI*, *ppk*), and cofactor/vitamin metabolism (*hemB*, *hemH*, and *hemL*) (Fig. 4A,B). Many of these upregulated genes are functionally linked to plant growth promotion. For

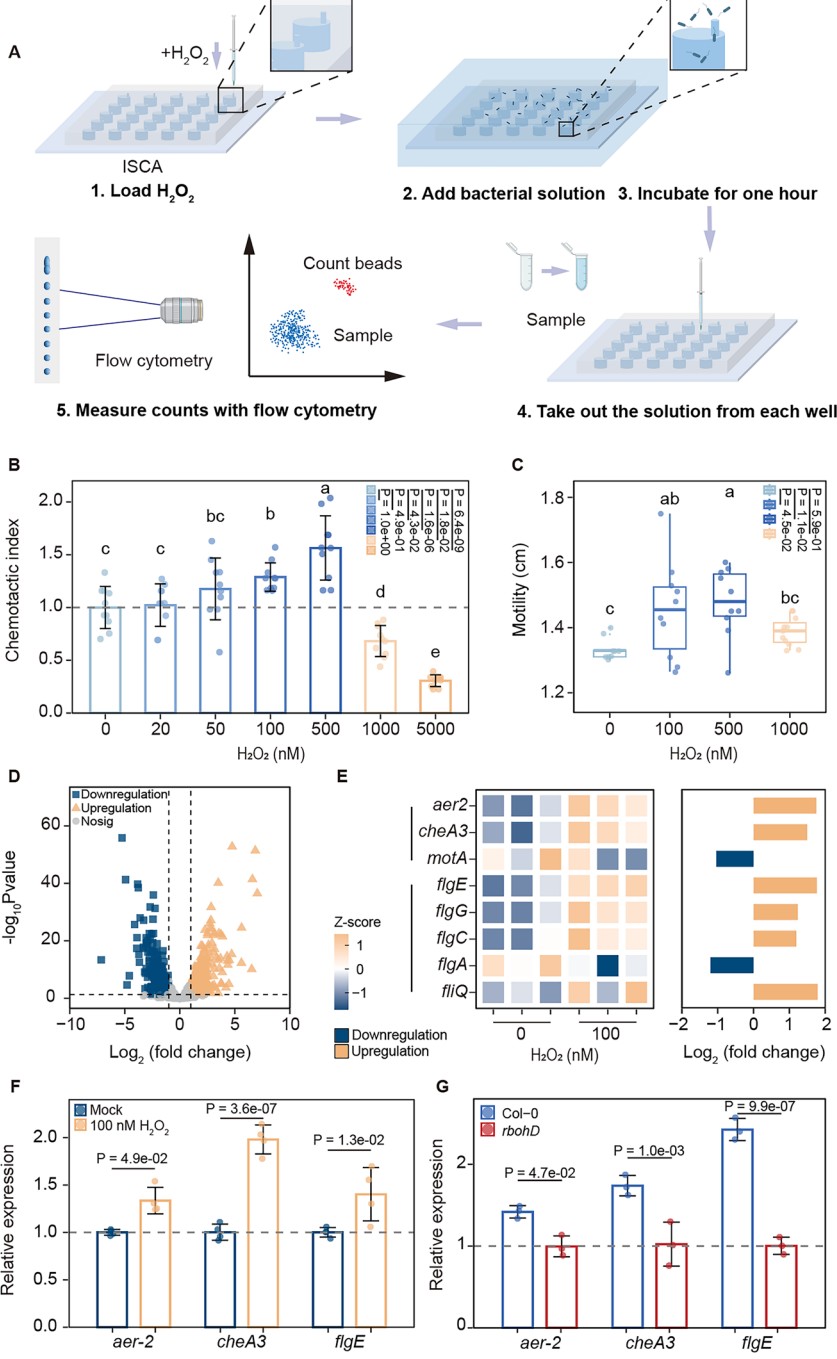

Figure 3. *Pseudomonas anguilliseptica* exhibits chemotaxis toward $H_2O_2$ (100 nM).

(A) Schematic diagram of the in situ chemotaxis assay (ISCA). (B) Chemotaxis assay of *P. anguilliseptica* toward $H_2O_2$. The chemotaxis index denotes the ratio of the number of bacteria in the wells supplemented with $H_2O_2$ to that in the control well ($n = 10$ biological replicates). Different letters indicate significant differences among the groups. P values are determined (ANOVA and Tukey test). (C) Effect of $H_2O_2$ on *P. anguilliseptica* motility ($n = 10$ biological replicates). Different letters indicate significant differences among the groups. P values are determined (ANOVA and Tukey test). (D) Volcano plot illustrating fold changes in gene expression in *P. anguilliseptica* under 100 nM $H_2O_2$ versus mock treatment conditions. Each condition included three biological replicates, and each biological sample represented the average of three technical replicates. Differential expression analysis was performed using edgeR. Group comparisons were conducted using a likelihood ratio test, and differentially expressed genes were identified at a false discovery rate (FDR) <0.05 with |log$_2$FC|>1. E Differential regulation of chemotaxis and motility-related genes in *P. anguilliseptica* following treatment with 100 nM $H_2O_2$ for 1 h. Gene expression normalized by Z-score is shown in the heatmap, while the bar plot illustrates fold changes, with yellow representing upregulation and blue indicating downregulation following $H_2O_2$ treatment ($n = 3$ biological replicates). (F) Effect of 100 nM $H_2O_2$ on gene expression associated with chemotaxis and motility. Gene expression was detected by qRT-PCR and normalized to that of *rpoA*. Data were the mean ± SD ($n = 4$ biological replicates). P values are determined (ANOVA and Tukey test). (G) Effect of *P. anguilliseptica* on the expression of chemotaxis and motility-related genes in the rhizosphere of Col-0 and *rbohD*. Gene expression was detected by qRT-PCR and normalized to that of *rpoA*. Data were the mean ± SD ($n = 3$ biological replicates). P values are determined (ANOVA and Tukey test). Source data are available online for this figure.

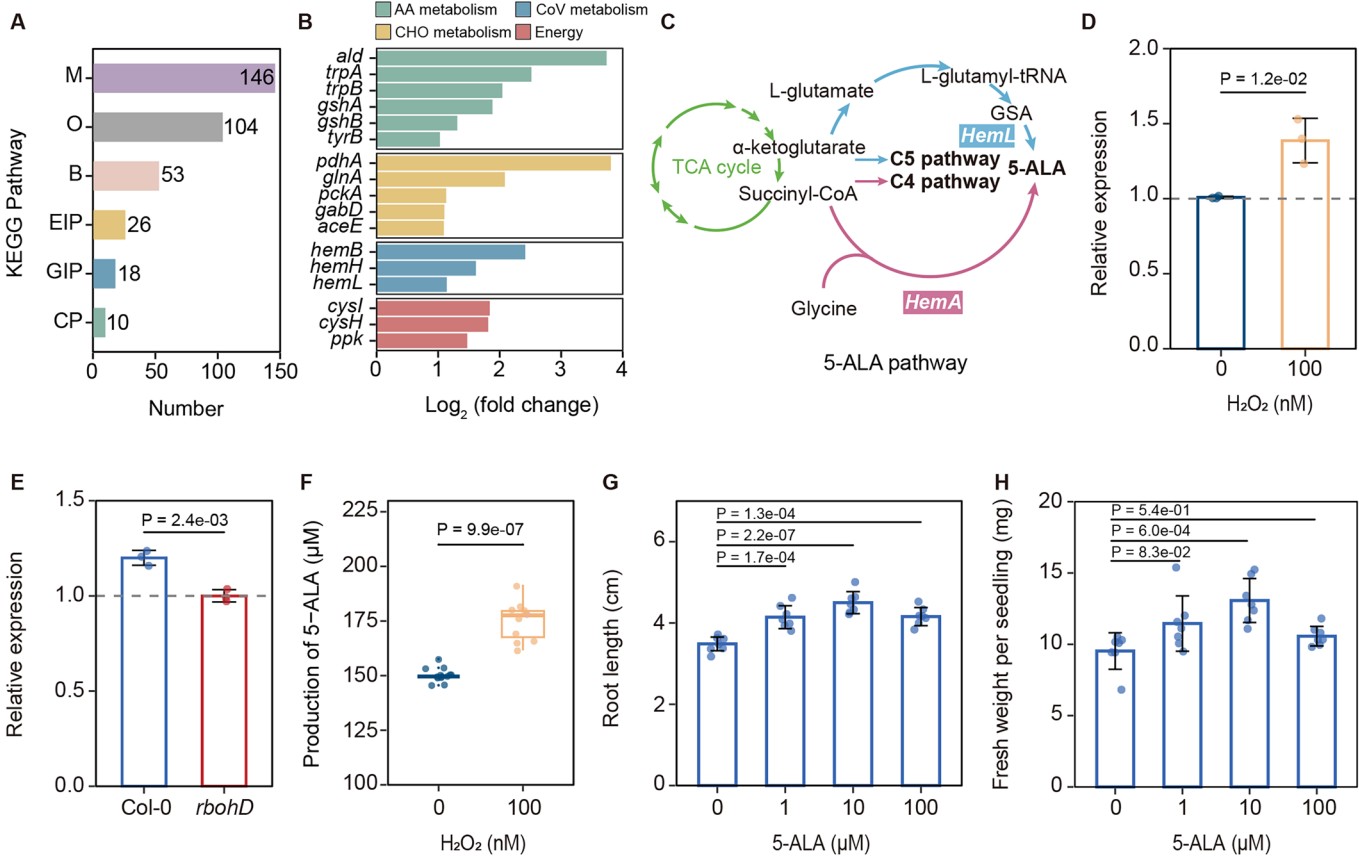

**Figure 4. H₂O₂ (100 nM) promote the synthesis of 5-aminolevulinic acid (5-ALA) in *Pseudomonas anguilliseptica*.**

(A) Functional classification of differentially expressed genes in *P. anguilliseptica* following H₂O₂ treatment. Differentially expressed genes were assigned to six KEGG BRITE hierarchies: Metabolism (M), Environmental Information Processing (EIP), Genetic Information Processing (GIP), Cellular Processes (CP), Brite hierarchies (B), and Others (O). (B) Summary of enriched metabolic pathways and key genes showing significant differential expression after 1 h treatment with 100 nM H₂O₂. Bar plots depict fold changes of upregulated genes following H₂O₂ treatment. AA amino acid, CoV coenzyme, CHO carbohydrate metabolism. ($n = 3$ biological replicates, Wilcoxon, $P \leq 0.05$). (C) Biosynthesis pathway of 5-aminolevulinic acid (5-ALA) in bacteria. (D) Effect of 100 nM H₂O₂ on gene expression associated with biosynthesis pathway of 5-ALA. Gene expression was detected by qRT-PCR and normalized to that of *rpoA*. Data were the mean ± SD ($n = 3$ biological replicates). *P* value is determined (ANOVA and Tukey test). (E) The relative expression of *hemL* in *P. anguilliseptica* within the rhizosphere of Col-0 and *rbohD*. Gene expression was detected by qRT-PCR and normalized to that of *rpoA*. Data were the mean ± SD ($n = 3$ biological replicates). *P* value is determined (ANOVA and Tukey test). (F) Levels of 5-ALA produced by *P. anguilliseptica*. Box plots display the median, upper, and lower quartile, with whiskers extending up to 1.5 times the interquartile range ($n = 11$ biological replicates). *P* value is determined (ANOVA and Tukey test). (G, H) Effect of 5-ALA on root length and seedling fresh weight. Data are presented as the mean ± SD ($n = 7$ biological replicates). *P* values are determined (ANOVA and Tukey test). Source data are available online for this figure.

instance: *trpA* and *trpB*, involved in tryptophan biosynthesis, may contribute to indole-3-acetic acid production, a phytohormone that promotes plant growth; *gshA* and *gshB*, encoding key enzymes in glutathione biosynthesis, may enhance bacterial oxidative stress tolerance and facilitate root colonization; *glnA*, responsible for glutamine synthesis, could enhance nitrogen availability in the rhizosphere; *cysI* and *cysH*, involved in sulfur metabolism, may influence microbial redox balance and fitness in planta; and *hemL* encoding glutamate-1-semialdehyde 2,1-aminomutase, an enzyme critical for the synthesis of 5-aminolevulinic acid (5-ALA) (Zhang et al, 2015), may enhance plant growth (Fig. 4C). The induction of *hemL* was confirmed by qRT-PCR (Fig. 4D). Consistent with ROS-dependent regulation, *hemL* transcript levels were significantly higher in *P. anguilliseptica* colonizing wild-type roots than in those colonizing *rbohD* mutants (Fig. 4E). Furthermore, we found that the addition of 100 nM H₂O₂ induced the production of 5-ALA in *P. anguilliseptica* (Fig. 4F), and exogenous application of 5-ALA

enhanced *Arabidopsis* root growth and seedling fresh weight (Fig. 4G,H). Together, these findings suggest that rhizosphere ROS not only facilitate the recruitment of *P. anguilliseptica* but also stimulate the production of growth-promoting metabolites, thereby contributing to plant growth.

## Immune elicitor-induced ROS recruit *P. anguilliseptica* toward the roots

We next examined whether *P. anguilliseptica* triggers ROS production in roots and whether this response depends on RBOHD. In wild-type plants, *P. anguilliseptica* inoculation induced a modest yet statistically significant ROS accumulation (~20% increase) at 1 hpi, as visualized by 3,3′-diaminobenzidine (DAB) staining (Fig. 5A–C). In contrast, *rbohD* mutants showed no detectable ROS induction under the same conditions (Fig. 5A–C). Notably, the magnitude of ROS accumulation was substantially

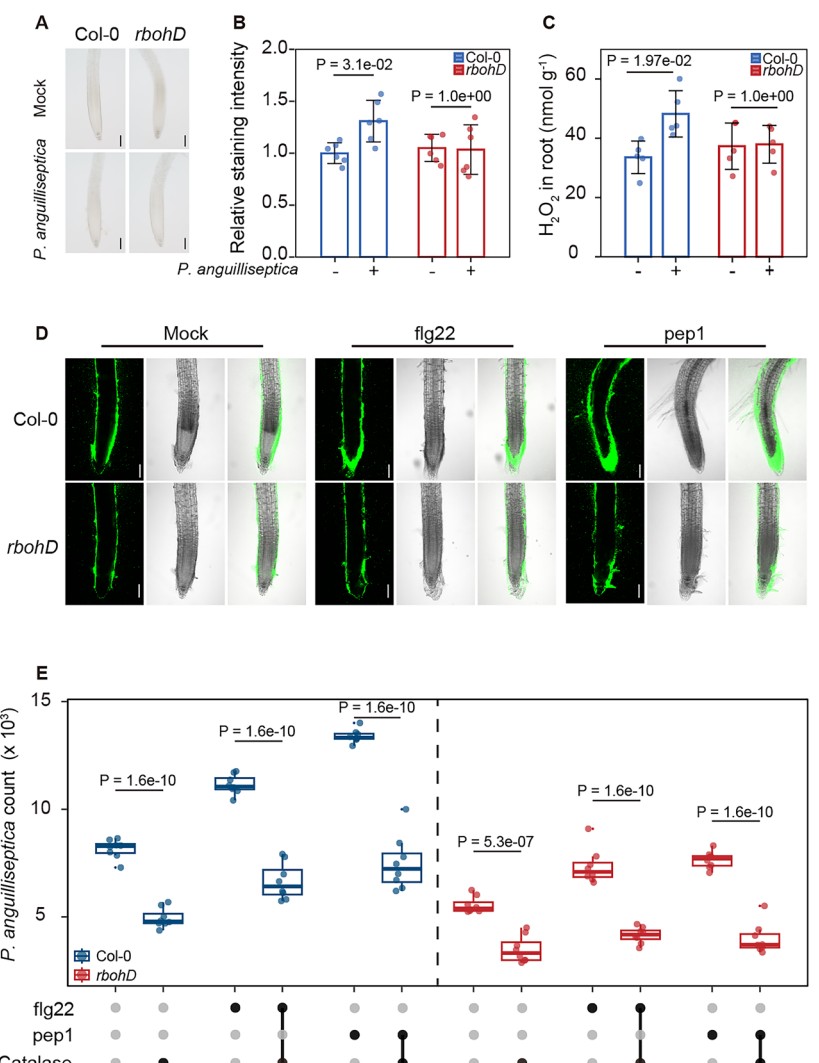

**Figure 5. Pretreatment with immune elicitors recruits *Pseudomonas anguilliseptica* toward the roots.**

(A–C) ROS levels in the roots. Five-day-old seedlings were inoculated with GFP-labeled *P. anguilliseptica* (OD$_{600}$ = 0.05). (A) Roots stained with 3,3′-diaminobenzidine (DAB). (B) Relative staining intensity quantified using ImageJ software. (C) H$_2$O$_2$ levels assayed by the ferrous-xylenol orange method. Scale bar, 50 μm. Data were means ± SD (n = 6 biological replicates in (B); 4 biological replicates in (C). *P* values are determined (ANOVA and Tukey test). (D, E) Flg22 and Pep1 pretreatment promotes rhizosphere colonization of GFP-labeled *P. anguilliseptica*, while ROS scavenging by catalase reduces this effect. Ten-day-old seedlings were inoculated with GFP-labeled *P. anguilliseptica* (OD$_{600}$ = 0.001). Representative images are shown in (D) and the bacterial population was quantified by flow cytometry at 3 dpi in (E). Scale bar, 100 μm. Box plots display the median, upper, and lower quartiles, with whiskers extending up to 1.5 times the interquartile range (n = 8 biological replicates). *P* values are determined (ANOVA and Tukey test). Source data are available online for this figure.

weaker than the robust responses typically elicited by pathogens or immune elicitors (>2-fold increase) (Entila et al, 2024; Torres et al, 2006). This prompted us to test whether immune elicitor-induced ROS could enhance *P. anguilliseptica* colonization in the rhizosphere. To address this, we pretreated the roots with the representative pathogen-associated molecular pattern flagellin peptide (flg22) and the representative damage-associated molecular pattern, plant elicitor peptide 1 (pep1) (Li et al, 2021), and then quantified GFP-labeled *P. anguilliseptica* in the rhizosphere. In wild-type plants, both flg22 and Pep1 treatments significantly increased *P. anguilliseptica* colonization in the rhizosphere by ~36.6 and 64.2%, respectively. While this increase was less pronounced in *rbohD* mutants, it remained observable, likely due to the redundant

role of RBOHF (Fig. 5D,E). Importantly, the addition of catalase strongly abolished the flg22- and pep1-induced enrichment of *P. anguilliseptica* (Fig. 5D,E). Together, these findings suggest that immune elicitor-triggered ROS may recruit *P. anguilliseptica* to the rhizosphere.

## Pathogen-induced ROS recruit *P. anguilliseptica* to mitigate disease severity

The levels of *P. anguilliseptica* in the rhizosphere following co-inoculation with *D. solani* were higher than those following co-inoculation with *E. sesbennia* (Fig. 2C), suggesting that *D. solani*-induced ROS production might contribute to recruiting *P.*

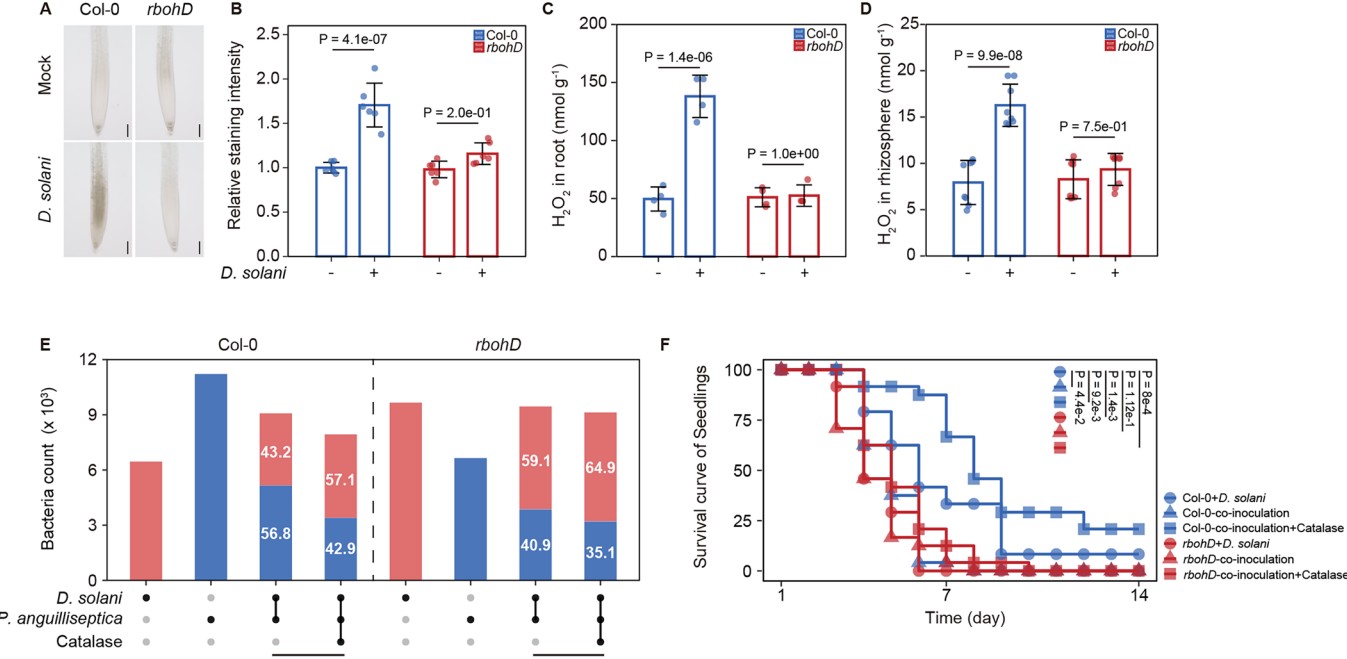

**Figure 6. Pathogen-induced ROS recruit *Pseudomonas anguilliseptica* to mitigate disease severity.**

(A–C) ROS levels within the roots following *D. solani* inoculation. Five-day-old seedlings were inoculated with *D. solani* (OD$_{600}$ = 0.025). ROS were detected using DAB staining (A, B) and the ferrous-xylenol orange method (C). Scale bar, 100 μm. Data were presented as the mean ± SD ($n$ = 6 biological replicates in (B), 4 biological replicates in (C). *P* values are determined (ANOVA and Tukey test). (D) ROS levels in the rhizosphere. *Arabidopsis* roots were grown within the Root Chip, following which the solution was harvested and the concentration of H$_2$O$_2$ was quantified using the Ampliflu Red method. Data were presented as the mean ± SD ($n$ = 8 biological replicates). *P* values are determined (ANOVA and Tukey test). (E) Root colonization competition between *P. anguilliseptica* and *D. solani*. Ten-day-old *Arabidopsis* seedlings were inoculated with *D. solani* (OD$_{600}$ = 0.001), either alone or co-inoculated with *P. anguilliseptica* at an equal cell density (OD$_{600}$ = 0.001) in the presence or absence of exogenous catalase. Bacterial populations were quantified by flow cytometry at 3 dpi. Colors indicate bacterial identity: blue, *P. anguilliseptica*; red, *D. solani*. Numbers show the relative percentage ($n$ = 6 biological replicates). (F) Disease assay induced by *D. solani*. Ten-day-old *Arabidopsis* seedlings were inoculated with *D. solani* (OD$_{600}$ = 0.001), either alone or co-inoculated with *P. anguilliseptica* in the presence or absence of exogenously applied catalase. Disease progression was monitored daily, and seedling mortality was scored based on visible leaf wilting ($n$ = 24 biological replicates). *P* values are determined (Wilcoxon).

anguilliseptica to the rhizosphere. To test this hypothesis, we quantified ROS levels in both roots and the rhizosphere following *D. solani* inoculation. A two- to three-fold increase in ROS was observed in wild-type roots after *D. solani* infection, but not in *rbohD* mutants (Fig. 6A–C). To assess H$_2$O$_2$ levels in the rhizosphere, we grew *Arabidopsis* seedlings in a previously developed Root Chip (Dai et al, 2022) (Fig. EV4A), in which rhizophere H$_2$O$_2$ was collected and measured. Although H$_2$O$_2$ concentrations were lower in the rhizosphere than in the roots, a three-fold induction was still observed after *D. solani* inoculation in the wild-type but not in *rbohD* plants (Fig. 6D). Notably, rhizosphere ROS levels triggered by *D. solani* were comparable to those in *apx1* and *cat2* mutants (Fig. EV4B). Furthermore, *P. anguilliseptica* exhibited significantly higher oxidative stress tolerance than *D. solani* in vitro (Appendix Fig. S6). Together, these results suggest that *D. solani*-induced ROS may also recruit *P. anguilliseptica* to the rhizosphere.

To test whether this ROS-mediated recruitment of *P. anguilliseptica* reduces *D. solani*-induced disease symptoms, we conducted co-inoculation experiments using seedlings. Consistent with our metagenomic sequencing data (Appendix Fig. S3), *rbohD* mutants harbored lower levels of *P. anguilliseptica* but higher levels of *D. solani* compared with wild-type plants, whereas catalase treatment diminished the genotype-dependent difference in *P. anguilliseptica*

abundance (Fig. 6E). Crucially, *P. anguilliseptica* co-inoculation significantly increased seedling survival through an ROS-dependent mechanism, as assessed by the Wilcoxon test (Fig. 6F). Soil-based co-inoculation experiments further showed that *P. anguilliseptica* alleviated *D. solani*-induced disease symptoms in wild-type plants but not in *rbohD* mutants (Fig. EV5A,B). Consistently, lower abundance of *P. anguilliseptica* was observed in the rhizosphere of *rbohD* mutants (Fig. EV5C). Collectively, these findings suggest that *D. solani*-induced ROS recruit beneficial *P. anguilliseptica*, which partially alleviates disease severity caused by *D. solani*.

## Discussion

Although ROS are generally considered oxidative stressors for microbes, their effects vary among specific microorganisms (Li et al, 2021; Song et al, 2021; Sahu et al, 2022). In this study, we found that ROS could induce *P. anguilliseptica* chemotaxis and promote the biosynthesis of growth-promoting compounds. Furthermore, pathogen pretreatment enhanced *P. anguilliseptica* abundance in the rhizosphere. These findings support a model in which plants produce ROS in root tissues as a defense response against pathogens, thereby suppressing pathogen colonization. Concurrently, root-derived ROS diffuse into the rhizosphere, where

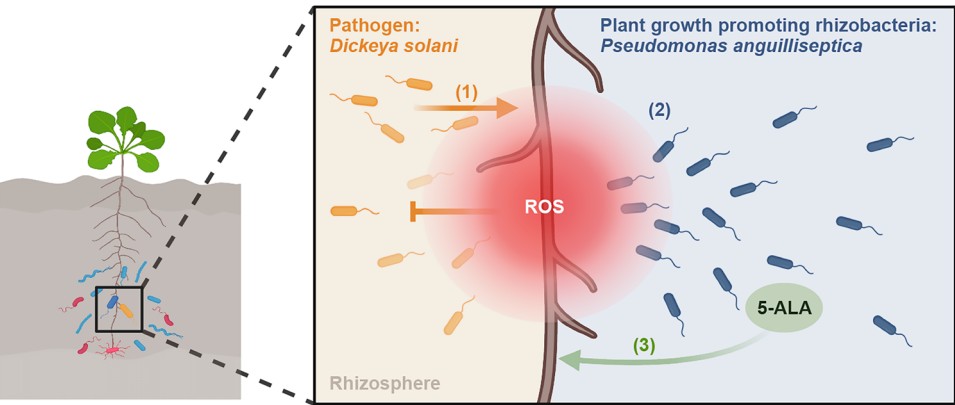

**Figure 7.  Schematic model illustrating that ROS recruits beneficial bacterium *Pseudomonas anguilliseptica* toward *Arabidopsis* roots.**

The rhizosphere is inhabited by various microorganisms. When the roots sense pathogens, ROS are produced in the roots to defend against pathogen colonization (1). Meanwhile, ROS diffuse into the rhizosphere, where they act as chemoattractants that attract the beneficial bacterium *P. anguilliseptica* toward the roots to mitigate disease severity (2). In addition, these ROS signals stimulate *P. anguilliseptica* to synthesize 5-aminolevulinic acid (5-ALA), which promotes plant growth (3).

they recruit the beneficial bacterium *P. anguilliseptica* toward the roots, ultimately promoting plant growth and enhancing plant resistance (Fig. 7). These insights pave the way for future research on the complex regulatory mechanisms underlying plant–microbe interactions and offer potential strategies for improving plant health through microbiome management.

ROS play important roles in plant defenses against pathogens (Gao et al, 2021; Wang et al, 2022). They are not only involved in direct antimicrobial activity, but also serve as crucial messengers that modulate immune responses, ensuring that plants can effectively recognize and counteract pathogens (Mittler et al, 2022; Wang et al, 2024). However, the roles of ROS in specific pathogens vary. For most pathogens, ROS are widely accepted to enhance plant cell wall strength through the cross-linking of cell wall components, thereby establishing a physical barrier against pathogen invasion (Fujita et al, 2020; Paauw et al, 2023). When pathogens infect through stomata, ROS trigger stomatal closure to restrict their entry (Kadota et al, 2014; Li et al, 2014). In addition, ROS serve as long-distance signals that initiate and propagate systemically acquired resistance, thereby triggering robust and widespread immune responses (Cao et al, 2024). Recently, novel roles for ROS have been reported; for instance, ROS inhibit the type II secretion system of *Xanthomonas*, effectively converting it into a beneficial microorganism (Pfeilmeier et al, 2024; Entila et al, 2024). Furthermore, ROS can induce auxin production in *Bacillus velezensis*, promoting its colonization in the host rhizosphere (Tzipilevich et al, 2021). Additionally, *Pseudomonas aeruginosa* exhibits chemotactic behavior toward antimicrobial compounds, including $H_2O_2$ (Oliveira et al, 2022). In this study, we found that $H_2O_2$ could promote the recruitment of *P. anguilliseptica*. Given the diversity of microorganisms, the role of ROS in plant–microbe interactions warrants further investigation.

Recent studies have highlighted the role of RBOHD-mediated ROS in maintaining homeostasis of the phyllosphere microbiota (Pfeilmeier et al, 2021). Consistent with this, we observed a significant difference in β-diversity within the root microbiota between the wild type and *rbohD* mutants. These results further support the notion that RBOHD-mediated ROS production

influences different microbial taxa, with significant alterations in the relative abundance of OTUs across various phyla. RBOHD is a member of the NADPH oxidase family, a group of enzymes that are highly conserved across various mammalian species (Sommer and Bäckhed, 2015), suggesting that the role of ROS in maintaining the microbiota might be conserved. Indeed, reduced ROS levels have been found to correlate with decreased gut microbiota diversity in mice (Herfindal et al, 2022). Furthermore, the reduction of NOX1-derived ROS indirectly decreases the abundance of *Lactobacilli* by removing their competitive advantage in the intestinal microenvironment (Matziouridou et al, 2018). Therefore, the regulation of microbial community structure, composition, and function by ROS appears to be conserved across multicellular organisms, highlighting their evolutionary significance.

In addition to their direct role in recruiting beneficial microorganisms, ROS may exert a profound influence on the metabolic pathways of plants and their associated microbiomes (Li et al, 2023; Sahu et al, 2022). This dynamic interplay could ultimately shape a unique and highly specialized microbial community tailored to a plant's evolving needs and environmental interactions (Deng et al, 2024). ROS can directly modulate central metabolic pathways like glycolysis and phenylpropanoid biosynthesis, which may selectively enrich beneficial microbes that enhance nutrient acquisition and prime defense responses (Pinheiro et al, 2010; Yaqoob et al, 2022). Additionally, different types of ROS may play different roles in specific microbes. Singlet oxygen can selectively kill gram-positive microbes, whereas superoxide anions exhibit higher toxicity to microbes than $H_2O_2$ (Wu et al, 2022). Here, we found that $H_2O_2$ induces *P. anguilliseptica* chemotaxis and the biosynthesis of growth-promoting compounds. However, we cannot rule out the possibility that $H_2O_2$-regulated metabolites or other types of ROS directly or indirectly influence the activity of *P. anguilliseptica*. Therefore, the molecular mechanism of ROS production underlying the enrichment of *P. anguilliseptica* in the rhizosphere requires further investigation.

In summary, our study reveals a previously overlooked role for rhizosphere ROS in orchestrating the selective recruitment of beneficial microbiota. This work not only enhances our mechanistic

understanding of plant–microbe interactions but also highlights the potential of using ROS-mediated microbial recruitment to engineer beneficial microbiomes. Ultimately, this strategy could enable the development of sustainable approaches to disease management and crop improvement, offering new opportunities to integrate plant immune regulation with ecological microbiome manipulation in agricultural systems.

# Methods

### Reagents and tools table

| Reagent/resource | Reference or source | Identifier or catalog number |
|---|---|---|
| **Experimental models** | | |
| *Arabidopsis:* wild-type, Columbia (Col-0) | ABRC | |
| *Arabidopsis: rbohD* (AT5G47910) | ABRC | CS9555 |
| *Arabidopsis: apx1-2* | ABRC | Salk_000249C |
| *Arabidopsis: cat2-1* | ABRC | Salk_091880 |
| *D. solani* | This paper | N/A |
| *P. dispersa* | This paper | N/A |
| *P. anguilliseptica* | This paper | N/A |
| *E. coli* DH5α | This paper | N/A |
| *E. sesbaniae* | This paper | N/A |
| **Recombinant DNA** | | |
| pBRG-GFP | This paper | N/A |
| **Antibodies** | | |
| **Oligonucleotides and other sequence-based reagents** | | |
| PCR primers | This study | Table EV2 |
| **Chemicals, enzymes and other reagents** | | |
| Flg22 | GenScript Biotech | N/A |
| Pep1 | GenScript Biotech | N/A |
| SYBR®Green Realtime PCR Master Mix | Vazyme | Cat# Q311-02 |
| RNAiso Plus | TaKaRa | Cat# 9109 |
| HiScript® II Q RT SuperMix for qPCR (+gDNA wiper) | Vazyme | Cat# R223-01 |
| Catalase | Aladdin | Cat#C163049-5g |
| H$_2$O$_2$ | Sigma-Aldrich | Cat# 88597-100ML-F |
| TIANgel Midi Purification Kit | TIANGEN | Cat# DP209-02 |
| TIANprep Mini Plasmid Kit | TIANGEN | Cat# DP103-02 |
| FastDNA™ SPIN Kit | MP Biomedicals | Cat# 6560200 |
| **Software** | | |
| ImageJ | GitHub | https://github.com/imagej/ImageJ |
| R | | https://www.r-project.org/ |
| USEARCH v11 | GitHub | https://github.com/rcedgar/usearch_old_binaries/ |
| MEGAHIT | GitHub | https://github.com/voutcn/megahit |

| Reagent/resource | Reference or catalog source | Identifier or catalog number |
|---|---|---|
| eggNOG-mapper | GitHub | https://github.com/eggnogdb/eggnog-mapper |
| DIAMOND | GitHub | https://github.com/bbuchfink/diamond |
| **Other** | | |
| BD FACS Melody | | |
| a NovaSeq X Plus PE150 | Illumina | |

## Plant materials and growth conditions

*Arabidopsis* mutants *rbohD* (CS9555), *apx1-2* (Salk_000249C), and *cat2-1* (Salk_091880) were obtained from the *Arabidopsis* Biological Resource Center (Ohio State University, USA) and used as previously described (Hong et al, 2023; Qi et al, 2023). Seeds were sterilized with 10% sodium hypochlorite and grown on 1/2 Murashige and Skoog (MS) agar medium containing 0.5% sucrose or in soil (Sun Gro Horticulture: Hawita professional = 1:1). Plants were cultivated in a growth chamber at 23 °C, 75% relative humidity, and a 12 h photoperiod.

For the seedling growth assay after bacterial inoculation, seedlings were transferred to sucrose-free agar medium. Inoculate the bacterial suspension at the root tip. For the seedling growth assay involving the addition of 5-ALA, seedlings were transferred to liquid 1/2 MS medium supplemented with 5-ALA. Root length and fresh weight were measured 5 d after incubation.

## 16S rRNA gene amplicon and metagenomic sequencing

Microbiome sequencing was performed as previously described with modifications (Bulgarelli et al, 2013). Briefly, rhizosphere soil was harvested from three-week-old *Arabidopsis* roots. Five plants were pooled into a single biological replicate. After gentle shaking to remove loosely adhering soil, the roots and tightly adhering soil aggregates were collected. Subsequently, phosphate-buffered saline (PBS) solution was added, and the mixture was vortexed to fully wash off rhizosphere microorganisms. DNA was extracted using a FastDNA™ SPIN Kit according to the manufacturer's instructions (MP Biomedicals, Solon, OH, USA) (Bulgarelli et al, 2012). Bacterial 16S rRNA genes were amplified using primers 515 F and 907 R, which target the V4-V5 regions. Amplified PCR products were sequenced using an Illumina HiSeq PE250 sequencing platform (Illumina, San Diego, CA, USA). Sequences were denoised into zOTUs using USEARCH v11 with a sequence similarity threshold of 0.97. Taxonomic annotations were performed using the Ribosomal Database Project training set v16 with the SINTAX algorithm (Cole et al, 2014) at a confidence threshold of 0.8.

Metagenomic shotgun sequencing was performed using a NovaSeq X Plus PE150 platform (Illumina, San Diego, CA, USA) at an average sequencing depth of 20 GB per sample. For functional annotation and analysis of metagenomic data, contigs were first assembled from clean reads using MEGAHIT (version 1.2.9), followed by alignment with the Kyoto Encyclopedia of Genes and Genomes database using eggNOG-mapper for functional annotation. Functional pathways were further annotated using multiple specialized databases, including NcycDB (Tu et al, 2019), PcycDB (Zeng et al, 2022), ScycDB (Yu et al, 2021), McycDB

(Qian et al, 2022), and VFDB (Dong et al, 2024). Annotation results were processed using the BLASTX option of DIAMOND v2.1.8 (Buchfink et al, 2021) with the following thresholds: amino acid sequence identity ≥60%, query length coverage ≥70%, and e-value ≤$10^{-5}$.

## Effect of $H_2O_2$ on bacterial growth, motility, and biofilm formation

*D. solani* and *P. dispersa* were cultured in Luria-Bertani medium. *P. anguilliseptica* was cultured on Reasoner's 2 A agar medium. *Ensifer sesbaniae* was cultivated on yeast mannitol agar. To evaluate the effect of $H_2O_2$ on bacterial growth, bacteria were cultured to the logarithmic phase. Resuspend bacteria in 10 mM $MgCl_2$ to an optical density ($OD_{600}$) of 1.0. Prepare agar containing varying concentrations of $H_2O_2$. A 5 μL aliquot was added to each treatment medium. Colony counts were recorded 2 d after incubation in the dark at 28 °C.

To assess the effect of $H_2O_2$ on bacterial motility, bacterial suspensions were treated with $H_2O_2$ and incubated for 6 h. Cells were then collected by centrifugation, resuspended in 10 mM $MgCl_2$, and 5 μL aliquots were spotted onto 0.3% agar plates. Incubate plates at 28 °C for 18 h. Measure colony diameter as an indicator of bacterial motility. To assess the impact of $H_2O_2$ on biofilm formation, bacteria were prepared as described above. Add corresponding concentrations of $H_2O_2$ to wells of a 96-well plate containing bacterial cultures. Incubate at 28 °C for 4 d. Quantify biofilm as previously described (Stepanović et al, 2000; Bhowmik, 2025).

## Generation of GFP-labeled bacterial strains

The GFP-labeled *D. solani* and GFP-labeled *P. anguilliseptica* strains were generated by introducing a modified plasmid (pBRG-GFP) (Zhu et al, 2022), carrying a kanamycin resistance gene and GFP fluorescent marker. Introduce the pBRG-GFP plasmid into competent cells *D. solani* and *P. anguilliseptica* using an electroporator with the following settings: V = 2.2 kV·cm$^{-1}$, C = 25 mF, R = 400 Ω.

## Bacterial inoculation

All bacterial strains were grown to logarithmic phase and washed once with 10 mM $MgCl_2$. For inoculation, cells were resuspended in 10 mM $MgCl_2$ to the desired concentrations ($OD_{600}$ = 0.001 for agar-based assays; $OD_{600}$ = 0.25 or 0.5 for soil drenching experiments). For agar plate-based assays, *Arabidopsis* seedlings were grown vertically on 1/2 MS medium containing 1% sucrose for 9 d. Seedlings with comparable root length were then transferred to sucrose-free 1/2 MS medium. For rhizosphere colonization assays, 1 μL of bacterial suspension ($OD_{600}$ = 0.001) was applied just below the root tip. GFP fluorescence was imaged 3 d after inoculation, following nine rinses with sterile water. For quantification, roots from four seedlings were pooled as one biological replicate, washed nine times with sterile water, vortexed for 35 s, and subjected to flow cytometry. For catalase treatments, catalase was applied at 1500 U/g, and its concentration was quantified using a BCA protein assay kit. BSA at the same protein concentration was used as a control.

For soil-based inoculation assays, seedlings were transplanted into soil and treated with *P. anguilliseptica* by soil drenching at one-, two-, and three-week post-transplantation ($OD_{600}$ = 0.5,

15 mL per plant, applied around but not directly on the root). Plants were phenotyped at 4 weeks for growth promotion. To quantify *P. anguilliseptica* levels in the rhizosphere, the solution containing bacteria surrounding plant roots were serially diluted and plated to calculate c.f.u. levels. For co-inoculation, GFP-labeled *P. anguilliseptica* and *D. solani* were mixed at equal cell densities and were applied at the same total volume ($OD_{600}$ = 0.5, 15 mL per plant). Disease symptoms were observed and recorded over time.

## RNA isolation and reverse transcription quantitative PCR (RT-qPCR)

Total RNA was extracted using an RNeasy kit according to the manufacturer's instructions (Tiangen Biotech, Beijing, China). cDNA was synthesized from 1 μg of isolated RNA using HiScript II reverse transcriptase (Vazyme Biotech, Nanjing, China). RT-qPCR was performed using SYBR Green Master Mix (Vazyme Biotech). The relative gene expression levels were calculated using the $2^{-\Delta\Delta Ct}$ method with 16S rRNA as an internal control (Paulin et al, 2009). All primers used for RT-qPCR are listed in Table EV2. To extract RNA from *P. anguilliseptica* in the root rhizosphere, bacteria adhering to the root surface were gently scraped off and immediately processed for RNA extraction.

## Bacterial RNA-Seq

Bacteria were cultured to logarithmic phase, and $H_2O_2$ was then added to a final concentration of 100 nM. Bacterial cells were collected by centrifugation after 1 h of incubation. cDNA libraries were constructed using a NovaSeq platform (Illumina, San Diego, CA, USA). The reference genome index was established in FASTA format, and filtered reads were aligned to the reference genome using Bowtie2. Gene read counts were calculated with HTSeq 0.6.1p2 (https://htseq.readthedocs.io/en/latest/) and was used as raw expression values. Differential expression analysis was performed using DESeq with the following criteria: log$_2$ fold-change >1 and differentially expressed genes were identified at a false discovery rate (FDR) <0.05.

## Detection of $H_2O_2$ levels in the roots and rhizosphere

To detect $H_2O_2$ levels in the roots, five-day-old *Arabidopsis* seedlings were transferred to sucrose-free 1/2 MS liquid medium and incubated overnight. Following treatment with *D. solani* ($OD_{600}$ = 0.025) and *P. anguilliseptica* ($OD_{600}$ = 0.05), the roots were washed three times with 10 mM $MgCl_2$. For $H_2O_2$ detection using DAB staining, the roots were immersed in 0.03% DAB in the dark for 8 min. Images were acquired using a stereomicroscope (Eclipse Ni-U; Nikon, Tokyo, Japan). The intensity of staining in each image was quantified using ImageJ software (https://imagej.nih.gov/ij/). For $H_2O_2$ detection via the ferrous-xylenol orange method, approximately 0.05 g of roots were ground in liquid nitrogen. After centrifugation at 14,000×$g$ at 4 °C, the supernatant was quantified as previously described (Wang et al, 2023). To detect $H_2O_2$ levels in the rhizosphere, *Arabidopsis* seeds were germinated in a 200 μL pipette tip inserted into the access hole of the Root Chip (Massalha et al, 2017). When the *Arabidopsis* root system penetrated approximately 1 cm into the hole, the rhizosphere fluid

was collected. $H_2O_2$ was quantified using the Ampliflu Red method (Dai et al, 2024).

## Chemotaxis assay

To measure the chemotactic response of microorganisms to $H_2O_2$, a microfluidics-based in situ chemotaxis assay was employed (Hallstrøm et al, 2022). Different concentrations of $H_2O_2$ were loaded into the wells of the microfluidic chip via micro-ports, allowing outward diffusion of $H_2O_2$ to generate a concentration gradient. Subsequently, the microfluidic chip was placed in 240 mL of bacterial suspension and incubate at 25 °C for 1 h. The solution in each well was harvested using a 1-mL syringe equipped with a 27-G needle. Bacterial levels were analyzed using a flow cytometer (BD FACS Melody with a 100-µm nozzle, USA). The forward scatter, side scatter, and GFP were recorded for each sample. Microbial populations were characterized based on side scatter and GFP fluorescence. The chemotaxis index (Ic) was determined as the ratio of cells in a specific treatment to those in the PBS control. Ic >1 indicates attraction, whereas Ic <1 indicates repulsion.

## Quantification of 5-ALA

The levels of 5-ALA secreted by bacteria were determined as previously described, with slight modifications (Li et al, 2014). Briefly, 400 µL of bacterial suspension was mixed with 200 µL sodium acetate buffer (pH 4.6) and 100 µL acetylacetone. The mixture was boiled for 15 min, and subsequently cooled to room temperature, followed by the addition of 700 µL Ehrlich's reagent under light-protected conditions. After 20 min of color development, absorbance was measured at 554 nm.

## Statistical analysis

Microbial community and functional analyses were conducted using the R software (version 4.2.3; R Core Team, Vienna, Austria). Community and functional gene matrices were normalized using the R package DESeq2 (Love et al, 2014). Principal coordinate analysis of the Weighted UniFrac (Hamady et al, 2010) distance was used to visualize microbial community variation between samples. The R package edgeR (Robinson et al, 2010) was used to identify differences in zOTU and functional genes between the two groups, with a false discovery rate less than 5%. LEfSe analysis was performed using the R package microbiomeMarker (Cao et al, 2022). Experimental data analysis and visualization were performed in R (version 4.2.3). No statistical tests were performed to determine sample sizes. Experiments were randomized, with plants shuffled weekly within the growth chambers throughout the study.

Graphics Figs. 2B, 3A, 7 and EV4A were created with BioRender.com.

## Data availability

The raw fastq files were deposited in the SRA under accession number PRJNA1215964.

The source data of this paper are collected in the following database record: biostudies:S-SCDT-10_1038-S44318-025-00685-w.

## Peer review information

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

## Acknowledgements

This work was supported by the National Natural Science Foundation of China (42090060 and 42277283), Key Research and Development Program of Zhejiang Province, China (2024SSYS0104, 2021C02064-7, and 2024ZD1000603), and the China Postdoctoral Science Foundation (2024M762854).

## Author contributions

**Xijie Guo**: Conceptualization; Data curation; Formal analysis; Validation; Investigation; Visualization; Methodology; Writing—original draft; Writing—review and editing. **Hengyi Dai**: Conceptualization; Data curation; Formal analysis; Validation; Investigation; Visualization; Methodology; Writing—original draft; Writing—review and editing. **Zhiyi Jia**: Visualization; Methodology; Writing—review and editing. **Ying Peng**: Validation; Visualization. **Luotian Lu**: Formal analysis. **Yaxing Su**: Formal analysis. **Jianwei Li**: Visualization; Methodology. **Qinghong Li**: Formal analysis. **Zeming Huang**: Formal analysis. **Yuchen Wang**: Formal analysis. **Fan Qi**: Visualization; Methodology. **Dayong Li**: Formal analysis. **Xiaofei Lv**: Methodology. **Yan Liang**: Conceptualization; Resources; Data curation; Formal analysis; Supervision; Funding acquisition; Validation; Investigation; Visualization; Methodology; Writing—original draft; Project administration; Writing—review and editing. **Bin Ma**: Conceptualization; Resources; Data curation; Formal analysis; Supervision; Funding acquisition; Validation; Investigation; Visualization; Methodology; Writing—original draft; Project administration; Writing—review and editing.

Source data underlying figure panels in this paper may have individual authorship assigned. Where available, figure panel/source data authorship is listed in the following database record: biostudies:S-SCDT-10_1038-S44318-025-00685-w.

## Disclosure and competing interests statement

The authors declare no competing interests.

# Expanded View Figures

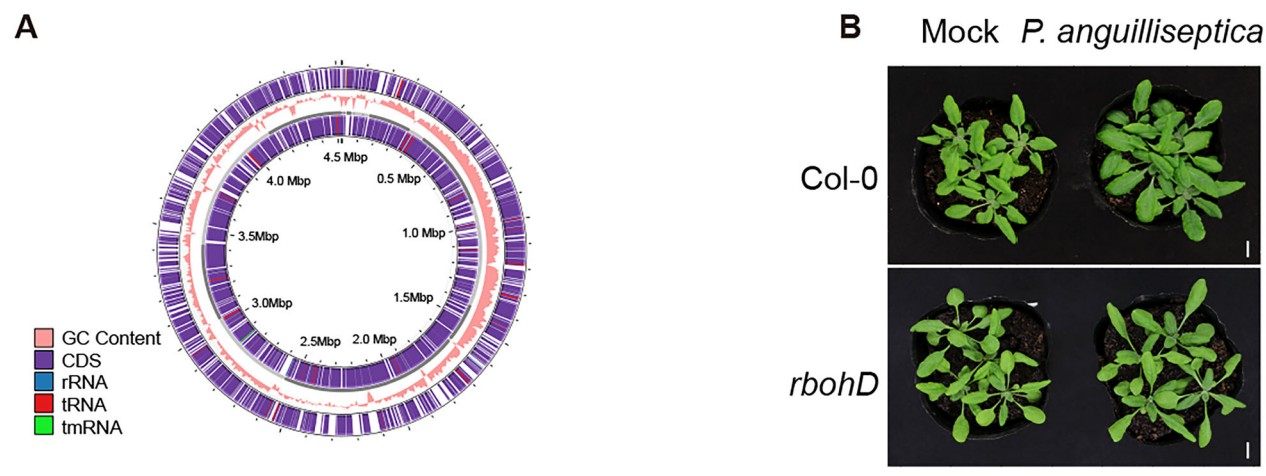

**A**

*Pseudomonas anguilliseptica*

GC Content
CDS
rRNA
tRNA
tmRNA

**B**   Mock   *P. anguilliseptica*

Col-0

*rbohD*

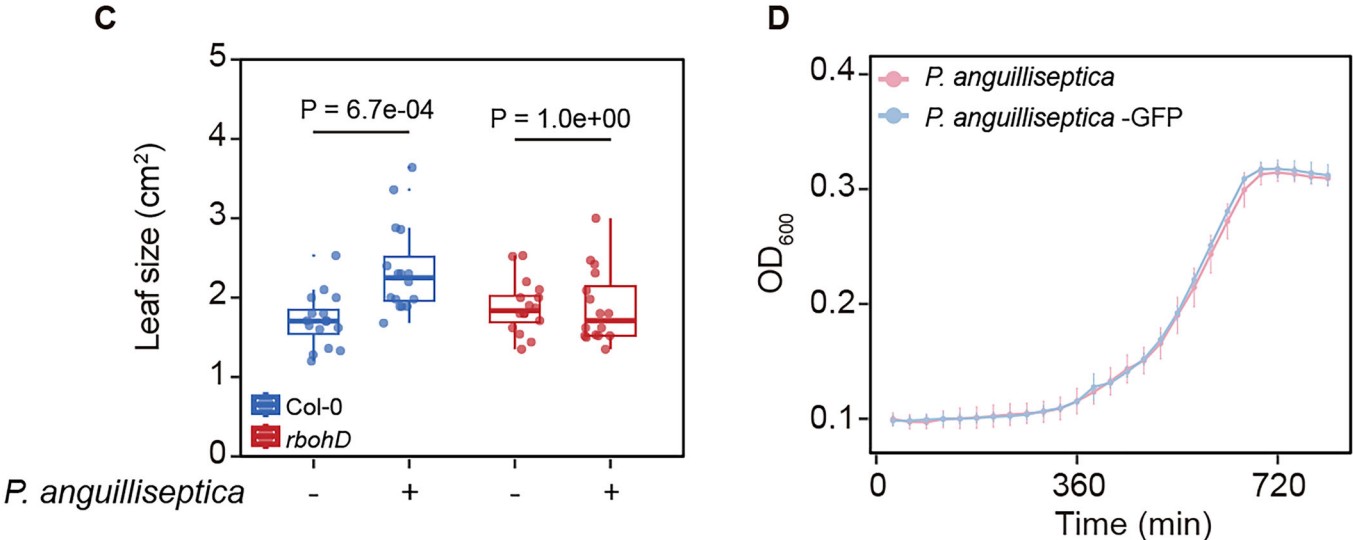

**C**

Leaf size (cm²)

P = 6.7e-04    P = 1.0e+00

Col-0
*rbohD*

*P. anguilliseptica*    −    +    −    +

**D**

$OD_{600}$

*P. anguilliseptica*
*P. anguilliseptica* -GFP

Time (min)

---

**Figure EV1.   *Pseudomonas anguilliseptica* promotes the growth of wild-type *Arabidopsis* but not *rbohD* mutants.**

(**A**) Circular map of the *P. anguilliseptica* genome. The complete genome is 4,523,881 bp in length. The outer and inner rings show coding sequences in purple, the middle ring indicates GC content in pink, tRNA genes are shown in blue, rRNA genes in red, and the tmRNA gene in green. (**B**, **C**) Soil-based inoculation with *P. anguilliseptica*. (**B**) Representative images; (**C**) Leaf size; Data represent mean ± SD ($n = 16$ biological replicates). *P* value is shown (ANOVA and Tukey test). Scale bar, 1 cm. (**D**) GFP-labeled *P. anguilliseptica* exhibits the same growth curve as *P. anguilliseptica*. Data were presented as the mean ± SD ($n = 3$ biological replicates). Source data are available online for this figure.

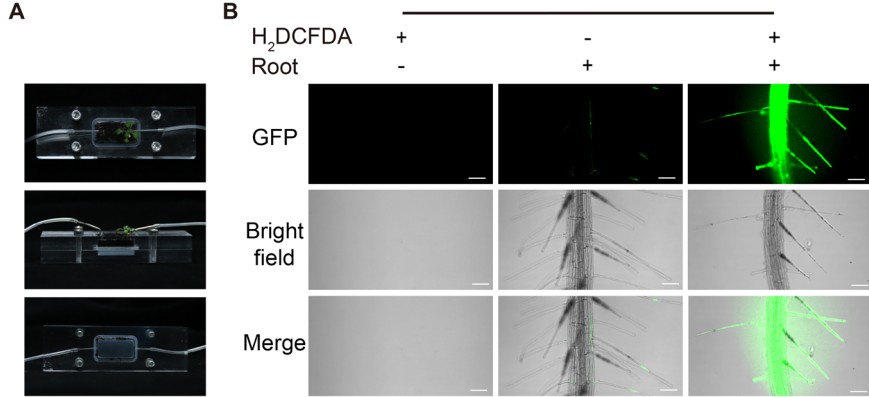

**Figure EV2. Microfluidic device for in situ visualization of rhizosphere ROS in *Arabidopsis*.**

(A) The microfluidic system used for in situ visualization of rhizosphere ROS. *Arabidopsis* seedlings were grown in soil within the device until roots fully enveloped the soil matrix. The bottom chip was then replaced with 0.8% agar containing 2′,7′-dichlorofluorescein (H₂DCFDA). After moistening the soil and roots, the root-soil interface was imprinted onto the agar for 1 h. Green fluorescence was observed. (B) Representative images. Scale bar, 100 μm. Source data are available online for this figure.

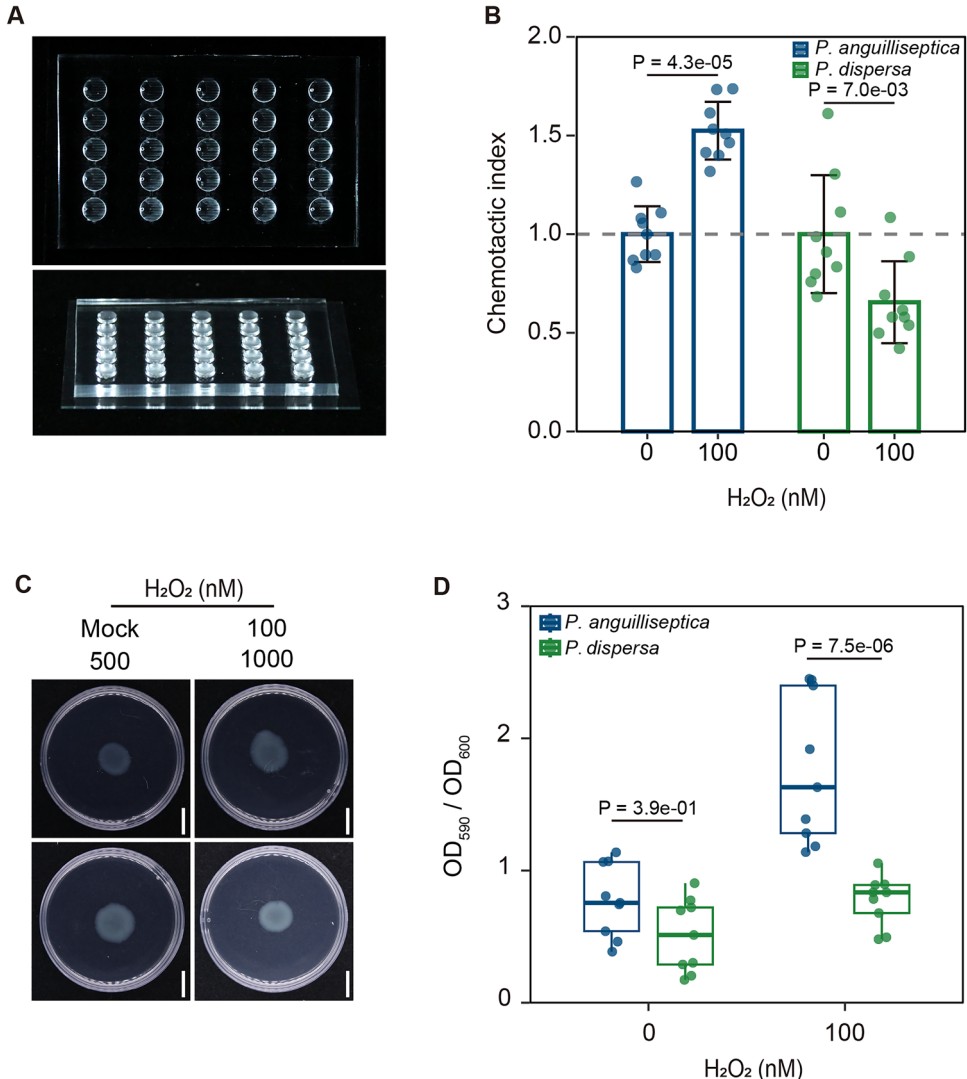

**Figure EV3.   Effect of 100 nM H₂O₂ on *Pseudomonas anguilliseptica.***

(**A**) Representative images of the chemotaxis chip. (**B**) Chemotaxis assay of *P. anguilliseptica* and *Pantoea dispersa* toward H₂O₂. The chemotaxis index denotes the ratio of the number of bacteria in the wells supplemented with H₂O₂ to that in the control well. Data were presented as the mean ± SD ($n = 9$ biological replicates). *P* values are determined (ANOVA and Tukey test). (**C**) Effect of H₂O₂ on *P. anguilliseptica* motility. Quantitative data were shown in Fig. 4. Scale bar, 1 cm. (**D**) Effect of H₂O₂ on biofilm formation by *P. anguilliseptica*. Box plots display the median, upper, and lower quartiles, with whiskers extending up to 1.5 times the interquartile range ($n = 9$ biological replicates). *P* values are determined (ANOVA and Tukey test). Source data are available online for this figure.

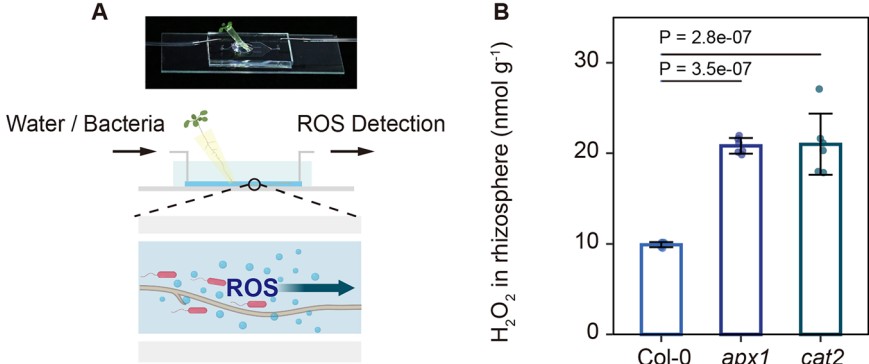

**Figure EV4. Measurement of rhizosphere ROS concentrations in Col-0, *cat2*, and *apx1* using a root-on-a-chip device.**

(A) Representative image of the root-on-a-chip system (top) and schematic illustration of ROS collection from the rhizosphere (bottom). (B) ROS concentrations in the rhizosphere. Rhizosphere ROS levels in ten-day-old Col-0, *apx1*, and *cat2* seedlings. Rhizosphere ROS levels were measured 45 min. Box plots display the median, upper, and lower quartiles, with whiskers extending up to 1.5 times the interquartile range ($n = 6$ biological replicates). $P$ values are determined (ANOVA and Tukey test). Source data are available online for this figure.

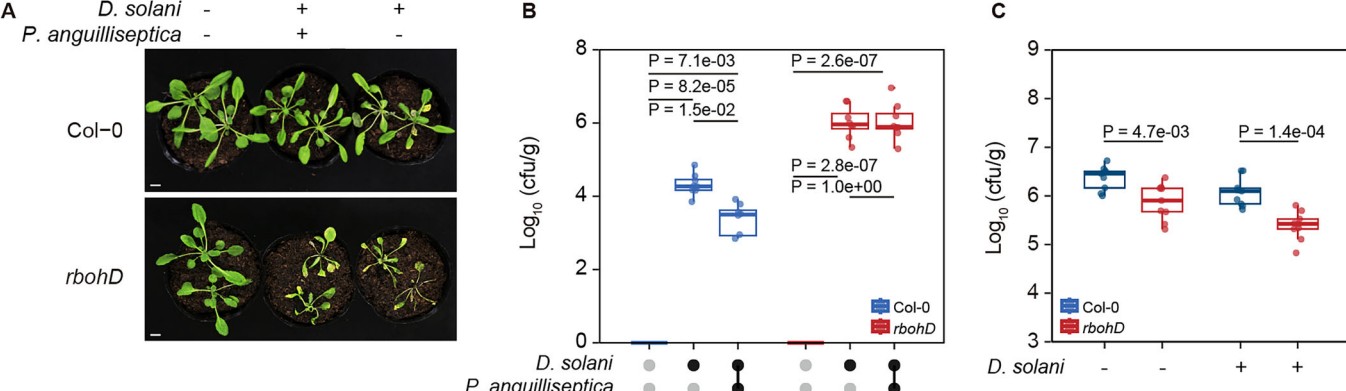

**Figure EV5. *Pseudomonas anguilliseptica* promotes plant resistance against *Dickeya solani* infection.**

(A, B) Three-week-old *Arabidopsis* plants were inoculated with either *D. solani* alone or co-inoculated with *P. anguilliseptica* ($OD_{600} = 0.5$, 15 mL). One-week post-inoculation, phenotypes were recorded (A), and *D. solani* colonization levels in rosette leaves were quantified (B). Scale bar, 1 cm. Data represent mean ± SD ($n = 8$ biological replicates). *P* values are determined (ANOVA and Tukey test). (C) Rhizosphere colonization of *P. anguilliseptica*. Three-week-old Col-0 and *rbohD* plants were inoculated with GFP-labeled *P. anguilliseptica* alone or co-inoculated with *D. solani* at equal cell densities ($OD_{600} = 0.5$, 15 mL per plant). One-week post-inoculation, rhizosphere soil was collected, and bacterial populations were determined by plating. Data represent mean ± SD ($n = 8$ biological replicates). *P* values are determined (Student's *t*-test). Source data are available online for this figure.

