## [Peer Review File · The EMBO Journal]

Reactive oxygen species in the rhizosphere orchestrate the recruitment of beneficial bacteria

Xijie Guo, Hengyi Dai, Zhiyi Jia, Ying Peng, Luotian Lu, Yaxing Su, Jianwei Li, Qinghong Li, Zeming Huang, Yuchen Wang, Fan Qi, Dayong Li, Xiaofei Lv, Yan Liang, and Bin Ma

Corresponding authors: Yan Liang (yanliang@zju.edu.cn) , Bin Ma (bma@zju.edu.cn)

Review Timeline:

Submission Date:	15th Oct 25
Editorial Decision:	23rd Nov 25
Revision Received:	10th Dec 25
Accepted:	18th Dec 25

Editor: Ieva Gailite

Transaction Report:

This manuscript was previously reviewed at another journal. As EMBO Press has a transfer agreement (including the identities of the referees) with that journal, revision was invited based on the reports from the previous journal.

Point-by-point responses to the reviewers' comments

We sincerely appreciate the reviewers' insightful comments, which have greatly helped us improve the manuscript. We have addressed most of the suggestions provided and incorporated the corresponding revisions, which are highlighted in yellow throughout the manuscript for ease of reference. Below, we provide a point-by-point response to the reviewers' comments and concerns (in blue).

Reviewers' Comments:

Reviewer #1 (Comments for the Author):

We thank you for your thoughtful suggestions and insights, which have enriched the manuscript and produced a better and more balanced account of the research.

[**Comment 1**] This manuscript proposes that reactive oxygen species (ROS) function as chemoattractants, recruiting the bacterium *Pseudomonas anguilliseptica*, which provides beneficial effects on plants, including root growth promotion and protection against the bacterial pathogen *Dickeya solani*. The authors profile rhizosphere bacterial communities and report that certain bacterial taxa are enriched (e.g., *D. solani*) and depleted (e.g., *P. anguilliseptica*) in *rbohD* mutant plants compared to wild-type plants. They demonstrate that *P. anguilliseptica* exhibits chemotaxis toward H₂O₂ *in vitro*. Transcriptomic analysis of *P. anguilliseptica in vitro* reveals that H₂O₂ induces the expression of *hemL* and promotes the production of 5-aminolevulinic acid (5-ALA), which, in turn, enhances root growth. The authors also show that *P. anguilliseptica* protects plants against *D. solani*. Based on these findings, they conclude that *P. anguilliseptica* is recruited via ROS-mediated chemotaxis, thereby promoting plant growth and enhancing resistance to pathogens.

ROS are widely recognized as primary defense molecules that act against pathogens either directly or through plant signaling. This study suggests an additional role for ROS in recruiting beneficial microbes that contribute to pathogen resistance. The proposed model is intriguing; however, the evidence presented is insufficient to support the authors' major conclusions, particularly those reflected in the title. There are two key issues. First, the evidence that chemotaxis toward ROS is responsible for enhanced bacterial colonization, root growth promotion, and pathogen resistance in wild-type plants compared to *rbohD* mutants is weak. Second, it remains unclear whether ROS directly influence *P. anguilliseptica* or its plant-mediated effects. Without addressing these critical issues, the underlying mechanism remains ambiguous, and the conclusions are not justified.

In summary, while the proposed model is conceptually appealing, it is not convincingly demonstrated.

Response: For the comment that “evidence that chemotaxis toward ROS is responsible for enhanced bacterial colonization, root growth promotion, and pathogen resistance in wild-type plants compared to *rbohD* mutants is weak”, we agree that the differences in *P. anguilliseptica* colonization, root growth promotion, and pathogen resistance between wild-type and *rbohD* plants were relatively modest. To address this, we conducted additional experiments with catalase treatment. Because catalase

attenuated these differences, the data suggest that chemotaxis toward ROS plays a key role in mediating these responses.

The concern that “it remains unclear whether ROS directly influence *P. anguilliseptica* or its plant-mediated effects” was addressed by incorporating new data, including transcriptome profiling of *P. anguilliseptica* following 100 nM H₂O₂ treatment, catalase treatment experiments assessing bacterial colonization and disease suppression, growth promotion assays in mature plants, spatial visualization of ROS in the rhizosphere, and gene expression analysis under plant-associated conditions. The key evidence supporting the role of ROS in recruiting beneficial microbes to enhance pathogen resistance includes:

(1) *In vitro* assays showing that *P. anguilliseptica* exhibits chemotaxis toward 100 nM H₂O₂, accompanied by upregulation of chemotaxis- and motility-related genes.

(2) Altered *P. anguilliseptica* colonization in *rbohD* mutant (weaker ROS response) and *cat2/apx1* (stronger ROS response) mutants compared to the wild type, with these differences attenuated by catalase treatment, implicating ROS as a critical *in vivo* recruitment signal.

(3) Suppression of *D. solani*-induced disease symptoms by *P. anguilliseptica*, an effect similarly reduced by catalase.

Together, these findings robustly support a model in which ROS function as chemoattractants for beneficial microbiota, thereby enhancing pathogen resistance.

[Comment 2] 1. Role of Chemotaxis in Colonization and Plant Benefits

The claim that *P. anguilliseptica*-mediated resistance against *D. solani* and growth promotion depend on chemotaxis toward ROS is not convincingly demonstrated. The authors show that *P. anguilliseptica* confers resistance independently of RBOHD, contradicting their hypothesis. Thus, the importance of chemotaxis in the observed phenotypes remains uncertain. To substantiate their claim, the authors should generate chemotaxis-deficient *P. anguilliseptica* mutant strains and assess their ability to colonize roots, promote plant growth, and confer pathogen resistance. Competitive colonization assays comparing wild-type and chemotaxis-deficient strains in wild-type and *rbohD* mutant plants would further clarify the role of chemotaxis.

Response: We sincerely appreciate the reviewer’s insightful suggestion regarding the generation of chemotaxis-deficient *P. anguilliseptica* mutants. In fact, we have made extensive efforts to establish a genetic manipulation system for this strain since August 2023, including attempts with homologous recombination (using

pK18mobsacB vectors), conjugation with donor strains (e.g., *Escherichia coli* S17 and *E. coli* WM3064), and CRISPR/Cas9-based genome editing. Despite optimizing selection markers, transformation conditions, and cell wall-weakening treatments, all attempts have so far been unsuccessful, suggesting that *P. anguilliseptica* is highly recalcitrant to conventional genetic manipulation. As an alternative [Comment 3], we employed catalase treatment to scavenge ROS and assess its impact on *P. anguilliseptica* colonization and disease suppression. These results are presented in Figure 6e and 6f of the revised manuscript (also see figure below), with the following description added:

line 217-225: “To test whether this ROS-mediated recruitment of *P. anguilliseptica* reduces *D. solani*-induced disease symptoms, we conducted co-inoculation experiments using seedlings. Consistent with our metagenomic sequencing data (Extended Data Fig. 3), *rbohD* mutants harbored lower levels of *P. anguilliseptica* but higher levels of *D. solani* compared with wild-type plants, whereas catalase treatment diminished the genotype-dependent difference in *P. anguilliseptica* abundance (Fig. 6e). Crucially, *P. anguilliseptica* co-inoculation significantly increased seedling survival through an ROS-dependent mechanism, as assessed by the Wilcoxon test (Fig. 6f). Soil-based co-inoculation experiments analyzed by one-way ANOVA further showed that *P. anguilliseptica* alleviated *D. solani*-induced disease symptoms in wild-type plants but not in *rbohD* mutants (Extended Data Fig. 11).”

We believe these data provide strong complementary evidence supporting the crucial role of chemotaxis in *P. anguilliseptica* colonization and associated plant benefits.

Fig.6. Pathogen-induced ROS recruit *Pseudomonas anguilliseptica* to mitigate disease severity. **e**, Root colonization competition between *P. anguilliseptica* and *D. solani*. Ten-day-old *Arabidopsis* seedlings were inoculated with *D. solani* (OD₆₀₀ = 0.001), either alone or co-inoculated with *P. anguilliseptica* at an equal cell density (OD₆₀₀ = 0.001) in the presence or absence of exogenous catalase. Bacterial populations were quantified by flow cytometry at 3 dpi. Colors indicate bacterial identity: blue, *P. anguilliseptica*; red, *D. solani*. Numbers show the relative percentage (n = 6). **f**, Disease assay survival curves of seedlings over time (day) for different genotypes and treatments. Survival is shown as a percentage of seedlings remaining healthy.

induced by *D. solani*. Ten-day-old *Arabidopsis* seedlings were inoculated with *Dickeya solani* ($OD_{600} = 0.001$), either alone or co-inoculated with *P. anguilliseptica* in the presence or absence of exogenously applied catalase. Disease progression was monitored daily, and seedling mortality was scored based on visible leaf wilting ($n = 24$). P values are shown (Wilcoxon).

Extended Data Fig. 11. *Pseudomonas anguilliseptica* promotes plant resistance against *Dickeya solani* infection. **a, b**, Three-week-old *Arabidopsis* plants were inoculated with either *D. solani* alone or co-inoculated with *P. anguilliseptica* ($OD_{600} = 0.5$, 15 mL). One week after inoculation, phenotypes were recorded (a), and *D. solani* colonization levels in rosette leaves were quantified (b). Scale bar, 1 cm. Data represent mean \pm SD ($n = 8$). Different letters indicate significant differences among the groups ($p < 0.05$, one-way ANOVA).

[Comment 3] 2. Potential Indirect Effects of ROS on Root Morphology

RBOHD-derived ROS influence not only directly microbial interactions but also root morphology, including lateral root development and lignification. The authors have not excluded the possibility that root morphological differences between wild-type and *rbohD* mutant plants contribute to the observed phenotypic differences. Additionally, the authors assert that "RBOHD-mediated ROS are produced in the apoplast, yet their diffusibility leads to a low-concentration gradient in the rhizosphere." However, it remains unclear whether a ROS gradient is established around roots and whether *P. anguilliseptica* exhibits enhanced chemotaxis toward wild-type plants relative to *rbohD* mutants. To address these issues, the authors should visualize the ROS gradient in the root environment and evaluate bacterial chemotaxis toward wild-type versus *rbohD* roots. Furthermore, ROS scavenger experiments could help differentiate direct bacterial responses from long-term plant-mediated effects.

Response: We sincerely appreciate the reviewer’s insightful suggestions. As RBOHD regulates lateral root emergence and lignin biosynthesis (Li, 2014 and Beata Orman-Ligeza, 2016), thereby potentially influencing root architecture, we agree that directly measuring ROS gradients in the root environment and employing ROS scavengers provides critical evidence for the role of ROS in recruiting *P. anguilliseptica*. These experiments further strengthen our conclusion by distinguishing ROS-mediated chemotaxis from indirect effects related to root morphological changes.

To visualize ROS gradients around roots, we employed a custom-designed microfluidic imaging device (please see figures below). The results are presented in Extended Data Fig. 6 of the revised manuscript, with corresponding descriptions added to the Results section:

line 121-127: “*Arabidopsis* seedlings were grown directly within the soil-filled chamber of the chip. Once the roots had extended and fully embedded themselves in the surrounding soil, the bottom layer of the chamber was carefully replaced with a solid agar gel filled with 2',7'-dichlorofluorescein (H₂DCFDA), a ROS sensitive fluorescent dye. After a 15-minute incubation period and subsequent removal of the culture layer. Green fluorescence was subsequently observed at the former root position, with gradient signals extending into the rhizosphere, indicating the diffusion of root-derived ROS into the rhizosphere environment (**Extended Data Fig. 6b**).”

Extended Data Fig. 6. Microfluidic device for *in situ* visualization of rhizosphere ROS in *Arabidopsis*. **a**, The microfluidic system used for *in situ* visualization of rhizosphere ROS. *Arabidopsis* seedlings were grown in soil within the device until roots fully enveloped the soil matrix. The bottom chip was then replaced with 0.8% agar containing 2',7'-dichlorofluorescein (H₂DCFDA). After moistening the soil and roots, the root-soil interface was

imprinted onto the agar for 1 hour. Green fluorescence was observed after the careful removal of the soil. **b**, Representative images. The left panel shows the root-present condition; the right panel shows the root-free control. Scale bar, 100 μm .

For ROS scavenging experiments, we employed catalase (H_2O_2 scavenger) to evaluate its effects on bacterial colonization and disease suppression (see Figure 2d, 2e, 5d, 5e, 6e, and 6f for colonization results). Our results demonstrated that exogenous catalase significantly reduced GFP-labeled *P. anguilliseptica* proliferation in the rhizosphere of wild-type, *rbohD*, *cat2*, and *apx1* mutants (Fig. 2d and 2e), eliminated the *flg22*- and *pep1*-induced enrichment of *P. anguilliseptica* (Fig. 5d and 5e), and attenuated *P. anguilliseptica*-mediated disease protection (Fig. 6e and 6f). These findings establish ROS as the primary determinant governing both recruitment of *P. anguilliseptica* and its plant-beneficial functions.

Fig. 2. RBOHD-derived ROS contribute to the enrichment of beneficial bacteria in the rhizosphere. **d, e**, Root colonization of GFP-labeled *P. anguilliseptica* in Col-0, *rbohD*, *apx1*, and *cat2*, with or without catalase treatment. Representative images are shown in (d) and bacterial populations were quantified by flow cytometry at 3 dpi (e). Scale bar, 100 μm . Box plots display the median, upper, and lower quartiles, with whiskers extending up to 1.5 times the interquartile range (n = 6). Different letters indicate significant differences among the groups (p < 0.05, one-way ANOVA).

Fig. 5. Pretreatment with immune elicitors recruits *Pseudomonas anguilliseptica* toward the roots. **d, e,** Flg22 and Pep1 pretreatment promotes rhizosphere colonization of GFP-labeled *P. anguilliseptica*, while ROS scavenging by catalase reduces this effect. Ten-day-old seedlings were inoculated with GFP-labeled *P. anguilliseptica* ($OD_{600} = 0.001$). Representative images are shown in (d) and the bacterial population was quantified by flow cytometry at 3 dpi in (e). Scale bar, 100 μm . Box plots display the median, upper, and lower quartiles, with whiskers extending up to 1.5 times the interquartile range ($n = 8$). Different letters indicate significant differences among the groups ($p < 0.05$, one-way ANOVA).

Fig.6. Pathogen-induced ROS recruit *P. anguilliseptica* to mitigate disease severity. **e,** Root colonization competition between *P. anguilliseptica* and *D. solani*. Ten-day-old *Arabidopsis* seedlings were inoculated with *D. solani* ($OD_{600} = 0.001$), either alone or co-inoculated with *P. anguilliseptica* at an

equal cell density ($OD_{600} = 0.001$) in the presence or absence of exogenous catalase. Bacterial populations were quantified by flow cytometry at 3 dpi. Colors indicate bacterial identity: blue, *P. anguilliseptica*; red, *D. solani*. Numbers show the relative percentage ($n = 6$). **f**, Disease assay induced by *D. solani*. Ten-day-old *Arabidopsis* seedlings were inoculated with *Dickeya solani* ($OD_{600} = 0.001$), either alone or co-inoculated with *P. anguilliseptica* in the presence or absence of exogenously applied catalase. Disease progression was monitored daily, and seedling mortality was scored based on visible leaf wilting ($n = 24$). P values are shown (Wilcoxon).

References:

1. **Li, N. et al.** *AtrbohD* and *AtrbohF* negatively regulate lateral root development by changing the localized accumulation of superoxide in primary roots of *Arabidopsis*. *Planta* **241**, 591-602 (2015).
2. **Orman-Ligeza, B. et al.** RBOH-mediated ROS production facilitates lateral root emergence in *Arabidopsis*. *Development dev*, **143**, 3328-3339 (2016).

[Comment 4] 3. Gene Expression Analysis in Plant-Associated Conditions

While the authors demonstrate induction of chemotaxis- and 5-ALA-related genes *in vitro*, they do not assess whether these genes are similarly induced in associated with wild-type versus *rbohD* mutant plants. Gene expression analyses in plants would provide critical insights into bacterial responses within a biologically relevant context. Additionally, the authors should show whether chemotaxis-related genes were upregulated in the *in vitro P. anguilliseptica* transcriptome analysis.

Response: We appreciate your suggestion to investigate the gene expression of chemotaxis- and 5-ALA-related genes in plant-associated conditions, particularly in the context of wild-type versus *rbohD* mutant backgrounds. In response, we isolated *P. anguilliseptica* from wild-type and *rbohD* mutant roots and performed qPCR to assess the expression of chemotaxis- and 5-ALA-related genes in *P. anguilliseptica*. Our results showed that bacteria colonizing *rbohD* roots showed significantly lower expression of chemotaxis-related genes (*aer2*, *cheA3*), motility-associated gene (*flgE*), and 5-ALA-related gene (*hemL*) than those from wild-type roots (please see figure Fig. 3g, 4e below). These findings support the notion that rhizosphere ROS facilitate *P. anguilliseptica* colonization *in vivo*.

Fig. 3. *Pseudomonas anguilliseptica* exhibits chemotaxis toward H₂O₂ (100 nM). g, Effect of *P. anguilliseptica* on the expression of chemotaxis and motility-related genes in the rhizosphere of Col-0 and *rbohD*. Gene expression was detected by qRT-PCR and normalized to that of *rpoA*. Data are the mean \pm SD (n = 3). Different letters indicate significant differences among the groups (p < 0.05, one-way ANOVA).

Fig. 4. H₂O₂ (100 nM) promote the synthesis of 5-aminolevulinic acid (5-ALA) in *Pseudomonas anguilliseptica*. e, The relative expression of *hemL* in *P. anguilliseptica* within the rhizosphere of Col-0 and *rbohD*. Gene expression was detected by qRT-PCR and normalized to that of *rpoA*. Data are the mean \pm SD (n = 3, Student's *t*-test, **p \leq 0.01).

Additionally, we refined our transcriptome analysis by optimizing *P. anguilliseptica* culture concentrations prior to H₂O₂ treatment. Our analysis demonstrated significant upregulation of gene clusters including chemotaxis-related genes (*aer2*, *cheA3*), motility-associated genes (*flgE*, *flgG*, *flgC*, *fliQ*) (please see Fig. 3d, 3e below), and plant growth-promoting metabolic pathways involving indole-3-acetic acid (IAA) biosynthesis, nitrogen/sulfur assimilation, and 5-aminolevulinic acid (5-ALA) production (please see Fig. 4a, 4b, 4c below). The corresponding descriptions have been incorporated into the revised manuscript:

line 151-155: “Furthermore, we performed transcriptomic analysis on bacteria treated with 100 nM H₂O₂, which revealed 358 upregulated and 314 downregulated genes at significant levels (**Fig. 3d**). Notably, key

chemotaxis-related genes (*aer2*, *cheA3*) and motility-associated genes (*flgE*, *flgG*, *flgC*, *fliQ*) were highly upregulated after H₂O₂ treatment (Fig. 3e).”

Fig. 3. *Pseudomonas anguilliseptica* exhibits chemotaxis toward H₂O₂ (100 nM). d, Volcano plot illustrating fold-changes in gene expression in *P. anguilliseptica* under 100 nM H₂O₂ versus mock treatment conditions. **e,** Differential regulation of chemotaxis and motility-related genes in *P. anguilliseptica* following treatment with 100 nM H₂O₂ for 1 h. Gene expression normalized by Z-score is shown in the heatmap, while the bar plot illustrates fold changes, with yellow representing upregulation and blue indicating downregulation following H₂O₂ treatment (n = 3).

line 164-176: “A total of 146 metabolism-related genes were upregulated following H₂O₂ treatment, with enrichment observed in pathways associated with amino acid metabolism (*ald*, *trpA*, *trpB*, *gshA*, *gshB*, *tyrB*), carbohydrate metabolism (*pdhA*, *glnA*, *pckA*, *gabD*, *aceE*), energy metabolism (*cysI*, *cysH*, *ppk*), and cofactor/vitamin metabolism (*hemB*, *hemH*, *hemL*) (Fig. 4a and 4b). Many of these upregulated genes are functionally linked to plant growth promotion. For instance: *trpA* and *trpB*, involved in tryptophan biosynthesis, may contribute to indole-3-acetic acid production, a phytohormone that promotes plant growth; *gshA* and *gshB*, encoding key enzymes in glutathione biosynthesis, may enhance bacterial oxidative stress tolerance and facilitate root colonization; *glnA*, responsible for glutamine synthesis, could enhance nitrogen availability in the rhizosphere; *cysI* and *cysH*, involved in sulfur metabolism, may influence microbial redox balance and fitness in planta; and *hemL* encoding glutamate-1-semialdehyde 2,1-aminomutase, an enzyme critical for the synthesis of 5-aminolevulinic acid (5-ALA)²⁷, may enhance plant growth (Fig. 4c).”

Fig. 4. H₂O₂ (100 nM) promote the synthesis of 5-aminolevulinic acid (5-ALA) in *Pseudomonas anguilliseptica*. **a**, Functional classification of differentially expressed genes in *P. anguilliseptica* following H₂O₂ treatment. Differentially expressed genes were assigned to six KEGG BRITE hierarchies: Metabolism (M), Environmental Information Processing (EIP), Genetic Information Processing (GIP), Cellular Processes (CP), Brite Hierarchies (B), and Others (O). **b**, Summary of enriched metabolic pathways and key genes showing significant differential expression after 1-hour treatment with 100 nM H₂O₂. Bar plots depict fold changes of upregulated genes following H₂O₂ treatment. AA, amino acid; CoV, coenzyme; CHO, carbohydrate metabolism. **c**, Biosynthesis pathway of 5-aminolevulinic acid (5-ALA) in bacteria.

References:

27. Zhang, J., Kang, Z., Chen, J. & Du, G. Optimization of the heme biosynthesis pathway for the production of 5-aminolevulinic acid in *Escherichia coli*. *Sci. Rep.*, **5**, 8584 (2015).

[Comment 5] 4. Ecological Relevance in Natural Settings

The physiological relevance of *P. anguilliseptica* recruitment in natural environments remains uncertain, especially given the relatively weak observed effects on plant growth and pathogen resistance. If *P. anguilliseptica* recruitment is not robust, its ecological significance may be limited. To strengthen their findings, the authors should assess whether *P. anguilliseptica*-mediated growth promotion and pathogen resistance depend on chemotaxis toward ROS under more natural conditions such as using soil environments or synthetic microbial communities.

Response: We sincerely appreciate your insightful comments regarding the ecological significance of *P. anguilliseptica* recruitment in natural environments. Addressing these concerns, we conducted soil-based inoculation assays to evaluate the impact of *P. anguilliseptica* on plant growth. Our results demonstrate that *P. anguilliseptica* significantly enhances leaf size, while no such effect was observed in *rbohD* mutants

(Please see Extended Data Fig. 5b and 5c below).

Extended Data Fig. 5. *Pseudomonas anguilliseptica* promotes growth of wild-type *Arabidopsis* but not *rbohD* mutants. b,c, Soil-based inoculation with *P. anguilliseptica*. (b) Representative images; (c) Leaf size; Scale bar, 1 cm. Data represent mean \pm SD (n = 16). Statistical analysis was performed using one-way ANOVA.

We also performed soil-based inoculation assays to test whether exogenous application of *P. anguilliseptica* could protect plants against *Dickeya solani* infection. Notably, the bacteria were inoculated around the roots rather than co-incubated with them. One week after co-inoculation, leaf tissues were homogenized and plated for bacterial quantification. The results showed that *P. anguilliseptica* significantly reduced *D. solani* colonization in wild type plants, whereas no protective effect was observed in the *rbohD* mutant (Please see Extended Data Fig. 11 below).

Extended Data Fig. 11. *Pseudomonas anguilliseptica* promotes plant resistance against *D. solani* infection. a, b, Three-week-old *Arabidopsis* plants were inoculated with either *D. solani* alone or co-inoculated with *P. anguilliseptica* (OD₆₀₀ = 0.5, 15 mL). One week after inoculation, phenotypes were recorded (a), and *D. solani* colonization levels in rosette leaves were quantified (b). Scale bar, 1 cm. Data represent mean \pm SD (n = 8). Different letters indicate significant differences among the groups (p < 0.05, one-way ANOVA).

[Comment 6] 5. Clarification of "Pathogen-Induced" ROS in the Title

The study does not demonstrate that pathogen-induced ROS are required for *P. anguilliseptica* chemotaxis. It is plausible that *P. anguilliseptica* chemotaxis is driven by constitutive or induced ROS production by commensal microbes rather than pathogen-specific ROS induction. The authors should demonstrate pathogens are required for *P. anguilliseptica* chemotaxis and *P. anguilliseptica*-mediated effects.

Response: Our data demonstrate that the levels of *P. anguilliseptica* in the rhizosphere were significantly higher following co-inoculation with *D. solani* compared to co-inoculation with *E. sesbannia*, a commensal microbe (Please see Fig. 2c below). To investigate whether *D. solani*-induced ROS are required for this recruitment, we performed co-inoculation experiments in the presence of catalase, a H₂O₂ scavenger. Notably, catalase treatment significantly suppressed *D. solani*-mediated recruitment of *P. anguilliseptica* to the rhizosphere (Please see response for [Comment 3] and Fig. 6e). In addition, catalase treatment significantly suppressed flg22- and pep1-mediated recruitment of *P. anguilliseptica* to the rhizosphere (Please see figure below). We believe that these findings support our proposed model, in which ROS triggered by pathogens serve as critical signaling molecules mediating the recruitment of beneficial microbes (Fig. 5d and 5e).

Fig. 2. RBOHD-derived ROS contribute to the enrichment of beneficial bacteria in the rhizosphere. c, Co-inoculation assay. Ten-day-old seedlings were co-inoculated with GFP-labeled *P. anguilliseptica* and either *D. solani* or *E. sesbaniae* at equal cell densities (OD₆₀₀ = 0.001). Bacterial populations were quantified by flow cytometry 3 days post-inoculation (dpi) (n = 9).

Fig.6. Pathogen-induced ROS recruit *Pseudomonas anguilliseptica* to mitigate disease severity. **e**, Root colonization competition between *P. anguilliseptica* and *D. solani*. Ten-day-old *Arabidopsis* seedlings were inoculated with *D. solani* (OD₆₀₀ = 0.001), either alone or co-inoculated with *P. anguilliseptica* at an equal cell density (OD₆₀₀ = 0.001) in the presence or absence of exogenous catalase. Bacterial populations were quantified by flow cytometry at 3 dpi. Colors indicate bacterial identity: blue, *P. anguilliseptica*; red, *D. solani*. Numbers show the relative percentage (n = 6).

Fig. 5. Pretreatment with immune elicitors recruits *Pseudomonas anguilliseptica* toward the roots. **d, e**, Flg22 and Pep1 pretreatment promotes rhizosphere colonization of GFP-labeled *P. anguilliseptica*, while ROS scavenging by catalase reduces this effect. Ten-day-old seedlings were inoculated with GFP-labeled *P. anguilliseptica* (OD₆₀₀ = 0.001). Representative

images are shown in (d) and the bacterial population was quantified by flow cytometry at 3 dpi in (e). Scale bar, 100 μm . Box plots display the median, upper, and lower quartiles, with whiskers extending up to 1.5 times the interquartile range ($n = 8$). Different letters indicate significant differences among the groups ($p < 0.05$, one-way ANOVA).

[Comment 7] 6. The authors emphasize that *P. anguilliseptica* exhibits chemotaxis toward "low" concentrations of ROS. How is "low" defined? Are ROS concentrations in *apx1* or *cat2* mutants—where *P. anguilliseptica* colonizes at higher levels than in wild-type plants—still considered low?

Response: We thank the reviewer for this comment. In our manuscript, H₂O₂ concentrations that directly inhibit bacterial growth (μM-mM range) were originally referred to as “high” (Huihui Zhang, 2021) whereas concentrations that do not markedly affect growth (nM range) were termed “low” (Hengyi Dai, 2022). To avoid ambiguity, we have now replaced these qualitative terms with precise quantitative values. In addition, we would like to clarify that directly comparing rhizosphere H₂O₂ levels (normalized per unit mass of root tissue) with the defined H₂O₂ concentrations (measured per unit volume) used in our *in vitro* chemotaxis assays presents technical challenges. Previous studies have demonstrated that H₂O₂ at infection or wounding sites can transiently accumulate to micromolar (μM) concentrations (Orozco-Cárdenas, 1999). While quantitative analyses have further confirmed the occurrence of H₂O₂ at comparable levels under diverse physiological and stress conditions (Sourav Chakraborty, 2016). Therefore, the 100 nM H₂O₂ concentration employed in our *in vitro* chemotaxis assays represents a relatively low physiological concentration. In the rhizosphere, our experiments, H₂O₂ concentrations in the rhizosphere of Col-0, *apx1*, and *cat2* were measured and found to be in the nM range. However, levels in *apx1* and *cat2* are higher relative to Col-0. We have also directly measured rhizosphere H₂O₂ concentrations in Col-0, *apx1*, and *cat2* to clarify these differences (Please see Extended Data Fig.9 below).

Extended Data Fig. 9. Measurement of rhizosphere ROS concentrations in Col-0, *cat2*, and *apx1* using a root-on-a-chip device. **a**, Representative image of the root-on-a-chip system (top) and schematic illustration of ROS collection from the rhizosphere (bottom). **b**, The root chip for measuring ROS concentrations in the rhizosphere. Rhizosphere ROS levels in ten-day-old Col-0, *rbohD*, *apx1*. Rhizosphere ROS levels were measured 45 minutes after

treatment. Box plots display the median, upper, and lower quartiles, with whiskers extending up to 1.5 times the interquartile range (n = 6, Student's *t*-test, ***p ≤ 0.001).

References:

1. **Zhang H, Liu Y, Wu G, Dong X, Xiong Q, Chen L, Xu Z, Feng H, Zhang R.** *Bacillus velezensis* tolerance to the induced oxidative stress in root colonization contributed by the two-component regulatory system sensor ResE. *Plant Cell Environ.*, **44**(9):3094-3102 (2021).
2. **Hengyi Dai, Binbin Wu, Baoliang Chen, Bin Ma, and Chiheng Chu.** Diel Fluctuation of Extracellular Reactive Oxygen Species Production in the Rhizosphere of Rice. *Environ. Sci. Technol.*, **56** (12), 9075-9082 (2022).
3. **Orozco-Cárdenas, C.A.Ryan.** Hydrogen peroxide acts as a second messenger for the induction of defense genes in tomato plants in response to wounding, systemin, and methyl jasmonate. *Proc. Natl. Acad. Sci. U.S.A.*, **96**(11), 6553-6557 (1999).
4. **Sourav Chakraborty, Amy L. Hill et al.** Quantification of hydrogen peroxide in plant tissues using Amplex Red. *Methods*, **109**, 105-113 (2016).

[Comment 8] 7. The statement, "It is widely accepted that beneficial microbes do not trigger significant ROS production in roots," lacks supporting references. Studies such as Chen et al. (2024, Cell Reports) and Tzipilevich et al. (2021, Cell Host & Microbe) demonstrate that beneficial microbes can induce ROS production, influencing root development and plant-microbe interactions.

Response: We sincerely appreciate the reviewer's careful evaluation of our manuscript and apologize for any confusion caused by our previous imprecise wording. We have revised the sentence:

line 185-186: "We next examined whether *P. anguilliseptica* triggers ROS production in roots and whether this response depends on RBOHD."

[Comment 9] 8. line 631 (page 28): The figure legend should use "*rbohD*" instead of "*RbohD*."

Response: We have revised the figure legend accordingly. (line 732)

[Comment 10] 9. The number of days post-inoculation for phenotype observation is not specified in the figure legends for Figures 2f, 5a, 5e, 6a, and Extended Data Figure 3e. Additionally, figure legends often lack sufficient detail to fully understand the experimental conditions. More comprehensive descriptions are needed.

Response: We appreciate this comment. We have carefully revised the legends for Figures 2d, 5a, 5d, 6a, and Extended Data Figure 3g (previously Figures 2f, 5a, 5e, 6a,

and Extended Data Figure 3e) to explicitly state the number of days post-inoculation (dpi) at which phenotypes were assessed, and to provide more detailed descriptions of the experimental conditions. For instance, in the rhizosphere colonization assay, *D. solani* and *P. anguilliseptica* were inoculated onto 10-day-old *Arabidopsis* seedlings, and colonization was quantified at 3 dpi. In addition, we have updated all relevant figure legends to provide more comprehensive descriptions of the experimental conditions, including plant genotype and age at inoculation, bacterial strain and concentration used, method of inoculation (e.g., soil drench), sample size (n), and statistical tests performed.

[Comment 11] 10. The x-axis units for Figures 6f and Extended Data Figure 3b are missing.

Response: We have added appropriate x-axis units to Figures 6f and Extended Data Figure 4c (previously Figures 6f and Extended Data Figure 3b).

Fig.6. Pathogen-induced ROS recruit *P. anguilliseptica* to mitigate disease severity. f. Disease assay induced by *D. solani*. Ten-day-old *Arabidopsis* seedlings were inoculated with *Dickeya solani* ($OD_{600} = 0.001$), either alone or co-inoculated with *P. anguilliseptica* in the presence or absence of exogenously applied catalase. Disease progression was monitored daily, and seedling mortality was scored based on visible leaf wilting (n = 24). P values are shown (Wilcoxon).

Extended Data Fig. 4. *rbohD* mutants exhibit increased susceptibility to

Dickya solani compared to the wild-type. a-e, Disease symptoms caused by *D. solani* in mature plants. Three-week-old plants in soil was inoculated with *D. solani*. The bacterial population in seedlings was enumerated by colony counting at 3 dpi. Scale bar, 1 cm. Box plots display the median, upper, and lower quartile, with whiskers extending up to 1.5 times the interquartile range (n = 13, Student's *t*-test, ****p* ≤ 0.001).

[Comment 12] 11. The y-axis label for Figure 3e appears incorrect.

Response: We thank the reviewer for noting this. The y-axis label has been corrected, and due to figure reordering the panel is now Fig. 3g (previously Fig. 3e).

Fig. 3. Pseudomonas anguilliseptica exhibits chemotaxis toward H₂O₂ (100 nM). g, Effect of *P. anguilliseptica* on the expression of chemotaxis and motility-related genes in the rhizosphere of Col-0 and *rbohD*. Gene expression was detected by qRT-PCR and normalized to that of *rpoA*. Data are the mean ± SD (n = 3). Different letters indicate significant differences among the groups (p < 0.05, one-way ANOVA).

[Comment 13] 12. line 58: Statistical support is required for the stated observation.

Response: We applied z-score normalization to depict relative enrichments or depletions of rhizosphere microorganisms at both the genus and phylum levels. This clarification has been incorporated into the manuscript:

line 58-66: “Z-score normalization of microbial abundance data revealed pronounced taxonomic shifts between wild-type and *rbohD* mutant groups at both the phylum and genus levels. At the phylum level, Actinomycetota, Bacillota, Bacteroidota, Bdellovibrionota, and Verrucomicrobiota were enriched, whereas Acidobacteriota, Cyanobacteriota, Myxococcota, Planctomycetota, and Pseudomonadota were depleted in *rbohD* mutants (Extended Data Fig. 1b). At the genus level, *Bradyrhizobium*, *Mesorhizobium*, *Mycobacterium*, *Rhizobium*, and *Streptomyces* were enriched, while *Caulobacter*, *Dokdonella_A*, *Parafilimonas*, *Phenylobacterium*,

Rhizomicrobium, and *Rhodanobacter* were depleted in *rbohD* mutants (Fig. 1e).”

[Comment 14] 13. line 77: The term "agricultural microbes" is ambiguous and should be clarified.

Response: We agree that the term “agricultural microbes” is ambiguous. We have now replaced it with “beneficial and pathogenic microbes” to more accurately describe the microbial groups under study. (line 80)

[Comment 15] 14. line 106: Figure 2d does not contain the referenced data.

Response: We have added the abundance information of the rhizosphere strains referenced in the manuscript for both *rbohD* and Col-0 plants (Supplementary Table 1). Additionally, we have included a reference describing *Ensifer sesbaniae* to provide further context. The updated citations are:

1. Wang, X.-L., M, K., Lin, Q.*et al.* Effect of ammonia-oxidizing bacterial strain that survives drought stress on corn compensatory growth upon post-drought rewatering. *Front. Plant Sci.*, **13**, 947476 (2022).

[Comment 16]15. line 160: Statistical support is required for this statement.

Response: We thank the reviewer for pointing this out. We have addressed this point in the revised manuscript:

line 195-200: “In wild-type plants, one-way ANOVA revealed that both flg22 and Pep1 treatments significantly increased *P. anguilliseptica* colonization in the rhizosphere by approximately 36.6% and 64.2%, respectively. While this increase was less pronounced in *rbohD* mutants, it remained observable, likely due to the redundant role of RBOHF (Fig. 5d, e). Importantly, the addition of catalase strongly abolished the flg22- and pep1-induced enrichment of *P. anguilliseptica* (Fig. 5d, e).”

[Comment 17]16. line 180: Statistical support is required for this statement.

Response: We have clarified our statement: the data in Fig. 6f were analyzed for statistical significance using the Wilcoxon test, confirming that disease severity differed significantly between treatments. In addition, we have supplemented the study with seedling inoculation experiments to further strengthen the conclusions. We have revised the sentence as follows:

line 222-225: “Crucially, *P. anguilliseptica* co-inoculation significantly increased seedling survival through an ROS-dependent mechanism, as assessed by the Wilcoxon test (Fig. 6f). Soil-based co-inoculation experiments analyzed

by one-way ANOVA further showed that *P. anguilliseptica* alleviated *D. solani*-induced disease symptoms in wild-type plants but not in *rbohD* mutants (Extended Data Fig. 11).”

Reviewer #2 (Comments for the Author):

We thank you for your thoughtful suggestions and insights, which have enriched the manuscript and produced a better and more balanced account of the research.

This paper reports a new role for ROS in the recruitment of beneficial microbes by roots. Starting from an amplicon-based microbiota comparison between wild-type and ROS-deficient mutants of *Arabidopsis*, the authors characterized few bacterial populations whose relative abundance decreased in ROS-deficient plants. Isolating a strain of *Pseudomonas* belonging to such populations, as well as a strain of *Dickeya* species enriched in ROS-deficient plants, they could confirm that pathogen-induced ROS led to enrichment in beneficial *Pseudomonas* and to mitigation of disease severity. Motility and chemotaxis were increased upon ROS exposure. Using *in vitro* exposure of the model *Pseudomonas* strain to oxidative stress, they noticed the elevated synthesis of 5-ALA, which, when applied pure, triggered root growth promotion. Whether this aspect was important in the defense against pathogen or in mediating the attraction by plants was not tested - it could have been done by generating a mutant in the gene encoding the enzyme responsible for 5-ALA synthesis, which would have brought a nice final touch to the story and a better link between ROS and recruitment.

This work is original and novel. It is significant, as it brings one more puzzle piece to the still largely unknown way how plants recruit beneficial microbes, especially when facing challenges (so-called “cry for help” hypothesis). I found the use of statistics and the robustness of the results convincing (but please systematically include the number of replicates and independent experiments in the legend). Previous work is appropriately credited. The MS is clearly written (few instances of overstatements or unclear passages are detailed below).

Suggestions for improvements in the text:

[Comment 1] L 30: please specify - the rhizosphere is an environment and cannot be referred to as genome (the authors probably mean root-associated bacteria).

Response: We have removed the term “rhizosphere” and revised the sentence to:

line 29-30: “The rhizosphere microbiome is often referred to as ‘second genome’ due to the indispensable role played by microbial communities in promoting plant growth and health¹⁴.”

Reference:

14. Berendsen, R. L., Pieterse, C. M. J. & Bakker, P. A. H. M. The rhizosphere microbiome and plant health. *Trends Plant Sci.*, **17**, 478-486 (2012).

[Comment 2] L32: please reformulate - it sounds like rhizosphere microbes only occupy ecological niches when plant encounter pathogens. But this occupancy occurs independently of or at least prior to pathogen presence, rather preventing the later establishment of pathogens than displacing them once they have reached the root.

Response: We agree that the original phrasing could be misleading. The sentence has been revised to:

line 30-35: “During pathogen challenge, plants activate immune responses that not only limit microbial invasion but also alter exudate composition, including polysaccharides, proteins, amino acids, organic acids, phytohormones, and phenolic compounds, which in turn facilitate the recruitment of beneficial microbes^{15,16}. Such beneficial microorganisms provide multilayered protection against pathogens through ecological niche competition, growth modulation, and immune response priming¹⁷⁻¹⁹.”

References:

15. Trivedi, P., Leach, J. E., Tringe, S. G., Sa, T. & Singh, B. K. Plant-microbiome interactions: from community assembly to plant health. *Nat. Rev. Microbiol.*, **18**, 607-621 (2020).

16. Afridi, M. S. et al. Harnessing root exudates for plant microbiome engineering and stress resistance in plants. *Microbiol. Res.*, **279**, 127564 (2024).

17. Van Elsas, J. D. et al. Microbial diversity determines the invasion of soil by a bacterial pathogen. *Proc. Natl. Acad. Sci. U.S.A.*, **109**, 1159-1164 (2012).

18. Liu, Y. et al. Beneficial microorganisms: Regulating growth and defense for plant welfare. *Plant Biotechnol. J.*, **23**, 986-998 (2025).

19. Zamioudis, C. & Pieterse, C. M. J. Modulation of host immunity by beneficial microbes. *Mol. Plant-Microbe Interact.*, **25**, 139-150 (2012).

[Comment 3] L41: please reformulate: most bacteria significantly increased their relative abundance (...) whereas the beneficial bacterium (...) notably decreased its relative abundance.

Response: We have revised the sentence to:

line 40-43: “Comparative analysis of rhizobacterial communities revealed significant compositional differences between wild-type and *rbohD* mutant plants, with the mutant rhizosphere showing altered colonization of pathogens

and beneficial rhizobacteria (e.g., *Dickeya solani* increased and *Pseudomonas anguilliseptica* decreased).”

[Comment 4] L44: did this enrichment happen also in mutant plants?

Response: Our results indicated that pathogen- and immune elicitor-induced *P. anguilliseptica* enrichment was significantly reduced, yet still detectable, in *rbohD* mutants compared to wild-type plants. Exogenous application of catalase (a ROS scavenger) strongly abolished *P. anguilliseptica* enrichment in both wild-type and mutant roots, underscoring the central role of ROS in this process. Given that RBOHF is known to function redundantly with RBOHD in *Arabidopsis* immunity, we propose that the residual enrichment observed in *rbohD* mutants may be attributable to RBOHF activity. In response to the reviewer's comment regarding the original line 44, we have removed results-specific descriptions from the objective section and added the following to the Results section:

line 197-200: “While this increase was less pronounced in *rbohD* mutants, it remained observable, likely due to the redundant role of RBOHF (**Fig. 5d, e**). Importantly, the addition of catalase strongly abolished the flg22- and pep1-induced enrichment of *P. anguilliseptica* (**Fig. 5d, e**).”

[Comment 5] L49: the title is an overstatement, please reformulate (the last sentence of this paragraph “contributes to the assembly” corresponds better to reality).

Response: We have revised the title as

line 48: “The *rbohD* mutants exhibit altered rhizosphere microbiome assembly.”

[Comment 6] L60: the two plant genotypes (to make it clearer)

Response: We have removed the phrase “two plant genotypes” and revised the sentence to improve clarity in the text:

line 58-66: “Z-score normalization of microbial abundance data revealed pronounced taxonomic shifts between wild-type and *rbohD* mutant groups at both the phylum and genus levels. At the phylum level, Actinomycetota, Bacillota, Bacteroidota, Bdellovibrionota, and Verrucomicrobiota were enriched, whereas Acidobacteriota, Cyanobacteriota, Myxococcota, Planctomycetota, and Pseudomonadota were depleted in *rbohD* mutants (**Extended Data Fig. 1b**). At the genus level, *Bradyrhizobium*, *Mesorhizobium*, *Mycobacterium*, *Rhizobium*, and *Streptomyces* were enriched, while *Caulobacter*, *Dokdonella_A*, *Parafilimonas*, *Phenylobacterium*,

Rhizomicrobium, and *Rhodanobacter* were depleted in *rbohD* mutants (Fig. 1e).”

[**Comment 7**] L68-70: this network structure and changes in interconnectivity are described but not interpreted nor discussed. Please either discuss them or remove them.

Response: We have added a detailed interpretation as:

line 70-73: “Species-level co-occurrence networks constructed from metagenomic profiles revealed markedly greater complexity in the *rbohD* rhizosphere, with 7,837 nodes and 158,934 edges compared to 5,386 nodes and 101,645 edges in the wild type (**Extended Data Fig. 1d**).”

[**Comment 8**] L77: attempted; please remove “agricultural”, or reformulate (“microbes with functions of relevance for agronomy or something similar” or simply “plant-associated bacteria”)

Response: We agree that the term “agricultural microbes” is ambiguous. We have now replaced it with “beneficial and pathogenic microbes” to more accurately describe the microbial groups under study. (line 80)

[**Comment 9**] L94: rhizosphere or root vicinity/environment

Response: We have deleted “root” (line 98) and carefully corrected similar instances throughout the revised manuscript.

[**Comment 10**] L98: caused L98

Response: We revised “causes” to “caused”. (line 101)

[**Comment 11**] L99: any reason why the authors focused on this particular strain species? Was any of the ulterior tests performed with other members of the bacterial populations enriched in wt compared with mutant plants?

Response: Among the eight bacterial species showing significantly reduced abundance in *rbohD* mutants, only *Pseudomonas anguilliseptica* was successfully identified from the ~100 isolated strains. The revised manuscript now states:

line 100-106: “To further explore the potential reasons why RBOHD dysfunction caused a decline in beneficial rhizobacteria, we isolated and characterized a collection of bacterial strains from key genera such as *Bacillus*, *Clostridium*, *Pseudomonas*, *Niallia*, *Acinetobacter*, *Pantoea*, *Enterobacter*, and *Massilia*. Subsequent comparative analysis revealed that, among these isolates, only *P. anguilliseptica* was consistently identified as a member of the

beneficial rhizobacterial community that exhibited significantly reduced abundance in the *rbohD* mutant²⁴.”

References:

24. Alaa, F. M. Effectiveness of exopolysaccharides and biofilm forming plant growth promoting rhizobacteria on salinity tolerance of faba bean (*vicia faba* L.). *Afr. J. Microbiol. Res.*, **12**, 399-404 (2018).

[Comment 12] L100: “the draft genome” - which one? An already available one or a newly sequenced one?

Response: The draft genome of *P. anguilliseptica* used in this study was newly sequenced as part of this work. The whole-genome sequence has been deposited in NCBI under accession number PRJNA1215964, which has been added in the Data availability section (line 416). A circular genome map is provided in the revised Extended Data Fig. 5a.

Extended Data Fig. 5. *P. anguilliseptica* promotes growth of wild-type *Arabidopsis* but not *rbohD* mutants. a, Circular map of the *P. anguilliseptica* genome. The complete genome is 4,523,881 bp in length. The outer and inner rings show coding sequences in purple, the middle ring indicates GC content in pink, tRNA genes are shown in blue, rRNA genes in red, and the tmRNA gene in green.

[Comment 13] L103: remove “other” since *P. anguilliseptica* is not a pathogen itself

Response: We removed “Other” in the revised manuscript. (line 111)

[Comment 14] L106: a beneficial

Response: We have added “a” in the revised manuscript. (line 113)

[Comment 15] L130-131: it is very strange to have both motility and biofilm formation increased, which usually are negatively correlated. How do the authors

explain this? Markers of chemotaxis showed higher expression, was there any correlative evidence for higher biofilm, e.g. in the RNA-Seq dataset generated?

Response: We acknowledge that motility and biofilm formation are often inversely regulated in bacteria, particularly under high oxidative conditions. In our studies, the simultaneous increase in both traits may be explained by difference in the duration of H₂O₂ treatment. Specifically, our motility assays involved short-term exposure (2 h) to 100 nM H₂O₂, after which the bacteria were washed and assessed for swimming behavior. In contrast, biofilm formation was evaluated under continuous exposure to 100 nM H₂O₂ over four days in static hydroponic conditions. Thus, the two experiments differed in the duration of ROS exposure. It is plausible that low-dose, transient ROS stimulation enhances motility through chemotactic activation, while prolonged exposure gradually promotes surface attachment and biofilm development, possibly as a protective adaptation. We have clarified this point in the revised manuscript as:

line 149-151: “Consistent with this hypothesis, we observed that 100 nM H₂O₂ treatment significantly enhanced *P. anguilliseptica* motility after 2 h (**Fig. 3c**; **Extended Data Fig. 8c**), and subsequently promoted biofilm formation upon prolonged exposure (**Extended Data Fig. 8d**).”

Similarly, our RNA-seq analysis following treatment with 100 nM H₂O₂ showed enhanced regulation of chemotaxis-related genes, but no significant changes in biofilm-related gene expression.

[Comment 16] L159: why would *P. anguilliseptica* be enriched in the rhizosphere of *rbohD* mutants upon elicitor treatment if the mutants cannot produce ROS (as shown in fig. 6)? Would that indicate that ROS are only one of multiple signals emitted by pathogen-challenged plants, which then recruit beneficial bacteria?

Response: As suggested, one possible explanation for the enrichment of *P. anguilliseptica* in the rhizosphere of *rbohD* mutants following elicitor treatment is that additional signals may contribute to its recruitment. Alternatively, functional redundancy with RBOHF could compensate for ROS generation post-elicitation (Jorge Morales, 2016). To distinguish between these possibilities, we utilized catalase, an H₂O₂ scavenger. Our findings demonstrate that catalase treatment strongly abolished *P. anguilliseptica* enrichment in the rhizosphere. Thus, residual ROS production likely accounts for the enrichment of *P. anguilliseptica* in *rbohD* mutants upon elicitor treatment.

References:

1. Jorge Morales, Yasuhiro Kadota, Cyril Zipfel *et al.* The *Arabidopsis* NADPH

oxidases *RbohD* and *RbohF* display differential expression patterns and contributions during plant immunity. *J. Exp. Bot.*, **67**, 1663-1676 (2016).

[**Comment 17**] L209: was ROS-mediated recruitment shown for other bacteria before (in the literature); and did you test it on other plant-associated bacteria? Likewise, how widespread among root-associated bacteria is the synthesis of 5-ALA? Was it a common feature of those species that were more abundant in wt than in ROS-deprived plants? Or would this be more anecdotal? Generating a mutant in the respective gene to quantify the role of 5-ALA in recruitment and/or pathogen symptom mitigation would have been interesting.

Response: We thank you for raising these important points regarding the novelty and generality of ROS-mediated recruitment and 5-ALA synthesis. To our knowledge, our study is the first to report a chemotactic response of a root-associated bacterium to H₂O₂. We agree that examining this phenomenon in a broader range of microbial taxa would be valuable. In preliminary screening of over 100 bacterial isolates, we observed H₂O₂-directed chemotaxis in *P. anguilliseptica* and *Massilia* sp. However, *Massilia* sp. was not enriched in *rbohD* mutants, we opted not to include these data in the current manuscript. The synthesis of 5-ALA is a common metabolic feature among bacterial species; therefore, we tested *hemL* expression in *Massilia* in response to 100 nM H₂O₂. Our results indicated that *hemL* expression was not induced in *Massilia*. Unfortunately, because we do not have access to other species that were reduced in *rbohD* mutants, we could not determine whether H₂O₂-induced synthesis of 5-ALA is common phenomenon.

Regarding the generation of knockout *P. anguilliseptica* mutants in the 5-ALA biosynthetic pathway, we sincerely appreciate the reviewer's valuable comments. Despite multiple attempts using conjugation, homologous recombination, and CRISPR/Cas9 approaches, we were unsuccessful in creating these mutants. As genetic manipulation of *P. anguilliseptica* remains technically challenging, our current work has focused on elucidating the chemotactic mechanisms toward ROS. The molecular role of 5-ALA in bacterial recruitment and/or pathogen symptom mitigation will be an important direction for future studies.

[**Comment 18**] L226: "H₂O₂ can alter the composition of metabolites" - please clarify. Of which metabolites are we talking here and what was the mechanism leading to promotion of microbial growth?

Response: We thank the reviewer for this insightful comment. We have addressed this comment in the manuscript as follows:

line 269-271: "ROS can directly modulate central metabolic pathways like

glycolysis and phenylpropanoid biosynthesis which may selectively enrich beneficial microbes that enhance nutrient acquisition and prime defense responses^{48,49}.”

References:

48. **Pinheiro, C. H. D. J. et al.** Regulation of glycolysis and expression of glucose metabolism-related genes by reactive oxygen species in contracting skeletal muscle cells. *Free Radic. Biol. Med.*, **48**, 953-960 (2010).
49. **Yaqoob, U., Jan, N. et al.** Crosstalk between brassinosteroid signaling, ROS signaling and phenylpropanoid pathway during abiotic stress in plants: Does it exist? *Plant Stress* **4**, 100075 (2022).

[**Comment 19**] L230: of the growth-promoting compound 5-ALA (or were there other growth-promoting compounds showing higher production rate upon ROS exposure?).

Response: In the revised manuscript, we added the following details:

line 168-176: “Many of these upregulated genes are functionally linked to plant growth promotion. For instance: *trpA* and *trpB*, involved in tryptophan biosynthesis, may contribute to indole-3-acetic acid production, a phytohormone that promotes plant growth; *gshA* and *gshB*, encoding key enzymes in glutathione biosynthesis, may enhance bacterial oxidative stress tolerance and facilitate root colonization; *glnA*, responsible for glutamine synthesis, could enhance nitrogen availability in the rhizosphere; *cysI* and *cysH*, involved in sulfur metabolism, may influence microbial redox balance and fitness in planta; and *hemL* encoding glutamate-1-semialdehyde 2,1-aminomutase, an enzyme critical for the synthesis of 5-aminolevulinic acid (5-ALA)²⁷, may enhance plant growth (**Fig. 4c**).”

References:

27. **Zhang, J., Kang, Z., Chen, J. & Du, G.** Optimization of the heme biosynthesis pathway for the production of 5-aminolevulinic acid in *Escherichia coli*. *Sci. Rep.*, **5**, 8584 (2015).

[**Comment 20**] L236: poorly understood or unclear

Response: In the revised manuscript, we have deleted the perspective section.

Suggestions for improvements in the figures:

[**Comment 21**] Fig. 1 - the triangle size in (f) is not clearly distinguishable; genus (horizontal label of f)

Response: We have adjusted the triangle sizes in Figure 1f to enhance visual distinguishability across genera. We have revised the x-axis label to clearly indicate genus.

Fig. 1. RBOHD-derived ROS mediate the assembly of the rhizosphere microbiome. f, The differential taxa between Col-0 and *rbohD* at the genus level based on LDA Effect Size score. Each triangle represents a unique zOTU, with its corresponding phylum depicted in various colors, and the size of each triangle corresponds to the $-\log_{10}(P\text{-value})$ ($p < 0.01$, Wilcoxon).

[Comment 22] Fig. 2 - (a): are the columns replicate plants? Please mention this in the material and methods and include the number of replicates in each legend of each figure. I suggest to not use the same color code for different things (e.g. genotypes of plants, but also two bacterial genotypes as in c); (a): were functional categories (beneficial, commensal, pathogenic) based on species levels, or strains (reconstructed genome) level? Please specify.

Response: Yes, each column represents a single biological replicate. For each replicate, rhizosphere soil was pooled from five individual plants. We have added this clarification to the Materials and Methods section and included the number of biological replicates (n) in each relevant figure legend throughout the manuscript for consistency.

Figure 2c (color coding): We have revised the figure to use distinct and consistent color schemes for plant genotypes and bacterial strains across all panels to avoid confusion.

In addition, the functional categories (beneficial, commensal, pathogenic) have been specified at the species levels in the revised manuscript.

[Comment 23] L543: with equal cell densities rather than equal volumes

Response: We revised this sentence to:

line 615-616: “Ten-day-old seedlings were co-inoculated with GFP-labeled *P. anguilliseptica* and either *D. solani* or *E. sesbaniae* at equal cell densities

(OD₆₀₀ = 0.001).”

[Comment 24] L626: “is depicted using databases” is not a clear formulation, please clarify. I found the legend of this figure not clear enough, it was not sufficient for me to understand the figure, especially the difference between the first column (are blue and red square fold changes (or what is meant by “gene abundance” and the blue bars the gene abundance, unlike what is written in the legend??).

Response: Thank you for highlighting the ambiguity. We have reworded the figure legend as:

line 718-723: “Identification of key metabolic pathways in rhizosphere microorganisms that differ between wild-type (Col-0) and *rbohD* mutants, including those related to methane, nitrogen, phosphorus, sulfur metabolism, and virulence factors. The blue bar graph illustrates the relative abundance of genes in Col-0 and *rbohD*. Colored squares denote relative enrichment, with red indicating higher abundance in Col-0 and blue indicating higher abundance in *rbohD* (*p < 0.05, **p < 0.01, ***p < 0.001, Wilcoxon).”

Reviewer #3 (Comments for the Author):

We thank you for your thoughtful suggestions and insights, which have enriched the manuscript and produced a better and more balanced account of the research.

[**Comment 1**] The paper by Guo and co-workers on the role of plant produced ROS on the microbial community in general and the specific impact on the plant beneficial strain *Pseudomonas anguilliseptica* and the plant pathogenic strain *Dickeya solani*.

The paper has a general interest in the scientific community, dealing with the molecular mechanisms behind plant-microbe interactions in the rhizosphere. There is a wide range of methods used to approach the question, and while they are clearly described, and the data presentation sound, they are mainly done under single or dual inoculations and under *in vitro* conditions.

While the paper is generally well-written and easy to follow, it would have benefitted from some clear hypotheses, e.g. what specific taxa would you expect to increase due to loss of ROS production, based on literature knowledge? Furthermore, I have some major concerns on the interpretation of the data based on the experimental setup:

Response: We performed soil-based inoculation to determine the growth-promoting effect of *Pseudomonas anguilliseptica* and its protective role against *Dickeya solani*-induced disease symptoms. Furthermore, we carefully revised our writing to clarify the hypotheses:

line 87-89: “Increased abundance of *Xanthomonas* pathogens has been reported in the *rbohD* phyllosphere²¹, however, no such increase was found in the rhizosphere. Instead, we observed enrichment of *D. solani*, a major causal agent of soft rot and blackleg in several crops²².”

References:

21. Entila, F., Han, X., Mine, A., Schulze-Lefert, P. & Tsuda, K. Commensal lifestyle regulated by a negative feedback loop between *Arabidopsis* ROS and the bacterial T2SS. *Nat. Commun.*, **15**, 456 (2024).
22. Matilla, M. A., Monson, R. E. & Salmond, G. P. C. *Dickeya solani*. *Trends Microbiol.*, **31**, 1085-1086 (2023).

[**Comment 2**] 1) Is it verified that the Col-0 and *rbohD* mutant has the same root exudate profile despite the leak of ROS to the rhizosphere? Without this established, it will be difficult to interpret the findings only in the light of ROS production or not.

Response: We agree that distinct root exudate profiles between Col-0 and *rbohD* mutants likely contribute to the observed differences in their associated microbial communities. To directly test whether ROS is the primary factor, we applied catalase to scavenge H₂O₂ in the rhizosphere. Catalase treatment strongly eliminated *P. anguilliseptica* enrichment, demonstrating that ROS is the dominant driver of this specific microbial recruitment. Furthermore, *in vitro* assays revealed that *P. anguilliseptica* exhibits a chemotactic response to 100 nM H₂O₂, accompanied by upregulated expression of chemotaxis- and motility-related genes. Together, we believe that these findings robustly support our model wherein ROS serve as chemoattractants for *P. anguilliseptica* recruitment. Nevertheless, we agree that the distinct root exudate profiles in *rbohD* mutants may also contribute to the altered rhizosphere microbiome composition. To reflect this more precisely, we have revised the heading of the first Results section to:

line 48: “The *rbohD* mutants exhibit altered rhizosphere microbiome assembly.”

for a more precise description. We then focused our investigation specifically on the role of ROS in mediating *P. anguilliseptica* recruitment.

[**Comment 3**] 2) The authors do not state that there are any differences in the plant parameters, e.g root length or fresh weight, when grown in soil. Hence, I assume that despite the higher levels of *D. solani*, this does not impact plant performance, independent on ROS production by the plant. The authors also find many more plant-beneficial microbes enriched in the *rbohD* mutant than in the WT Col-0. Could it be that there is not an overall difference in the microbiome performance under complex conditions, despite differences in specific taxa and genes (based on metagenome analysis)?

Response: We compared the growth phenotypes of wild-type and *rbohD* mutant plants cultivated in soil. While no significant growth differences were observed between wild-type and *rbohD* mutants, we cannot exclude the possibility that subtle physiological variations might influence their respective rhizosphere microbiome. As noted by the reviewer, both wild-type and *rbohD* mutants grow well in natural soil despite the presence of *D. solani*. Importantly, the pathogen levels in natural soil are relatively low compared to disease conditions. Indeed, *rbohD* mutants exhibited more severe disease symptoms upon *D. solani* inoculation (please see Extended Data Fig. 4a below).

Extended Data Fig. 4. *rbohD* mutants exhibit increased susceptibility to *Dickya solani* compared to the wild-type. a-e, Disease symptoms caused by *D. solani* in mature plants. Three-week-old plants in soil was inoculated with *D. solani*. The bacterial population in seedlings was enumerated by colony counting at 3 dpi. Scale bar, 1 cm. Box plots display the median, upper, and lower quartile, with whiskers extending up to 1.5 times the interquartile range (n = 13, Student's *t*-test, ****p* ≤ 0.001).

To determine whether the observed reduction in plant-beneficial microbes in *rbohD* mutants represents a common microbiome phenotype under complex environmental conditions, we examined the literature for other immunity-deficient mutants, such as *npr1* and *cpr5* (Pfeilmeier, S, 2022). While significant shifts in the abundance of plant-beneficial microbes were detected in *rbohD* mutant phyllosphere, *npr1* and *cpr5* mutants did not exhibit the same changes. Although our study focused on rhizosphere microbes, these findings suggest that the disruption of RBOHD specifically leads to a decline in certain plant-beneficial microbial populations.

Fig. 2 Effect of plant genotype on the leaf endosphere community. d, Relative abundance of phyla (and classes for Proteobacteria) of endosphere

bacteria on the indicated plant genotypes. Genotypes are ordered by decreasing abundance of Gammaproteobacteria. Asterisks (or hashtags for Firmicutes) denote significant differences in taxa on a genotype compared to Col-0 in a two-sided t-test ($P < 0.05$, Benjamini-Hochberg adjusted, $n = 12$).

Reference:

1. Pfeilmeier, S., Petti, G.C., Bortfeld-Miller, M. *et al.* The plant NADPH oxidase RBOHD is required for microbiota homeostasis in leaves. *Nat Microbiol*, **6**, 852-864 (2021).

[Comment 4] 3) In line 175-176 the authors use the finding that *P. anguilliseptica* has a higher tolerance to H_2O_2 than *D. solani*, to support the proposed model that “plants generate ROS in their roots to suppress the colonization of *D. solani*, while the low concentrations of ROS in the rhizosphere recruit *P. anguilliseptica* toward the *Arabidopsis* roots.” However, the concentrations where inhibition H_2O_2 inhibition is observed was at 400 μM , while all other supporting data is done at a much lower concentration of 100 nM. The authors need to comment on this, in relation to their conclusions.

Response: While our *in vitro* experiments showed bacterial inhibition at 400 μM H_2O_2 , we do not expect *in vivo* H_2O_2 levels to reach such high concentrations. It is worth noting that superoxide exhibits stronger bactericidal effects than H_2O_2 *in vivo*. The recruitment of *P. anguilliseptica* to the root by rhizosphere H_2O_2 raises an intriguing question: Why does root ROS not impose oxidative stress on this bacterium? To address this, we tested *P. anguilliseptica*'s oxidative stress tolerance *in vitro* and confirmed its remarkable resistance. However, since this manuscript primarily investigates the chemotactic function of rhizosphere H_2O_2 (at ~100 nM) in recruiting *P. anguilliseptica* to the root zone, we have deleted speculative discussions regarding the potential antibacterial effects of root ROS.

[Comment 5] 4) The presented importance of the study for potential applications in sustainable agriculture is somewhat overinterpreted as the majority of data are obtained under *in vitro* conditions in assays with only 1-2 bacterial strains.

Response: In the revised manuscript, we have removed the speculative discussion regarding potential applications in sustainable agriculture. However, to maintain relevance to agricultural contexts, we conducted soil-based inoculation assays to evaluate the impact of *P. anguilliseptica* on plant growth. Our results show that *P. anguilliseptica* significantly increased leaf size of wild-type plants, whereas no such effect was detected in *rbohD* mutants (please see Fig. 2a, Extended Data Fig. 5b-d below).

Fig. 2. RBOHD-derived ROS contribute to the enrichment of beneficial bacteria in the rhizosphere. **a**, *P. anguilliseptica*-induced enhancement of seedling fresh weight in Col-0, but not in *rbohD*. Box plots display the median, upper, and lower quartiles, with whiskers extending up to 1.5 times the interquartile range (n = 16).

Extended Data Fig. 5. *Pseudomonas anguilliseptica* promotes growth of wild-type *Arabidopsis* but not *rbohD* mutants. **b,c**, Soil-based inoculation with *P. anguilliseptica*. (b) Representative images; (c) Leaf size; Scale bar, 1 cm. Data represent mean \pm SD (n = 16). Statistical analysis was performed using one-way ANOVA. **d**, GFP-labeled *P. anguilliseptica* exhibits the same growth curve as *P. anguilliseptica*. Data are presented as the mean \pm SD (n = 3).

We also performed soil-based inoculation assays to test whether exogenous application of *P. anguilliseptica* could protect plants against *D. solani* infection. Notably, the bacteria were inoculated around the roots rather than co-incubated with them. One week after co-inoculation, leaf tissues were homogenized and plated for bacterial quantification. The results showed that *P. anguilliseptica* significantly reduced *D. solani* colonization in wild-type plants, whereas no protective effect was observed in the *rbohD* mutant (please see Extended Data Fig. 11 below). Therefore, we summarize the key findings and agricultural implications of our study as:

line 279-281: “In summary, our study reveals a previously underappreciated signaling role of rhizosphere ROS in orchestrating the selective recruitment of

beneficial microbiota. By linking the molecular mechanisms of plant immunity with the dynamics of microbial community assembly, these findings illuminate a chemical dialogue through which plants actively shape their root-associated microbiome. This work not only enhances our mechanistic understanding of plant-microbe interactions but also highlights the potential of harnessing ROS-mediated microbial recruitment as a strategy for microbiome engineering. Ultimately, leveraging these ROS-microbiota interactions could enable the development of sustainable approaches to disease management and crop improvement, offering new opportunities to integrate plant immune regulation with ecological microbiome manipulation in agricultural systems.”

Extended Data Fig. 11. *Pseudomonas anguilliseptica* promotes plant resistance against *D. solani* infection. a, b, Three-week-old *Arabidopsis* plants were inoculated with either *D. solani* alone or co-inoculated with *P. anguilliseptica* ($\text{OD}_{600} = 0.5$, 15 mL). One week after inoculation, phenotypes were recorded (a), and *D. solani* colonization levels in rosette leaves were quantified (b). Scale bar, 1 cm. Data represent mean \pm SD (n = 8). Different letters indicate significant differences among the groups (p < 0.05, one-way ANOVA).

Minor comments:

[Commen6] line 36-38: reference 19 is used to support the statement that RBOHD-mediated ROS production results in a low concentration gradient in the rhizosphere. While the reference describes such a concentration gradient, it does not link it to plant ROS production. Please take this into account in the rationale for your work.

Response: In the revised manuscript, we have deleted the description of the ROS gradient in the introduction. Instead, we visualized it using a custom-designed microfluidic imaging device (please see figure below). The results have been included as Extended Data Fig. 6 in the revised manuscript, with corresponding descriptions added to the Results section:

line 121-127: “*Arabidopsis* seedlings were grown directly within the soil-filled chamber of the chip. Once the roots had extended and fully embedded themselves in the surrounding soil, the bottom layer of the chamber was carefully replaced with a solid agar gel filled with 2',7'-dichlorofluorescein (H₂DCFDA), a ROS sensitive fluorescent dye. After a 15-minute incubation period and subsequent removal of the culture layer. Green fluorescence was subsequently observed at the former root position, with gradient signals extending into the rhizosphere, indicating the diffusion of root-derived ROS into the rhizosphere environment (Extended Data Fig. 6b).”

Extended Data Fig. 6. Microfluidic device for *in situ* visualization of rhizosphere ROS in *Arabidopsis*. **a**, The microfluidic system used for *in situ* visualization of rhizosphere ROS. *Arabidopsis* seedlings were grown in soil within the device until roots fully enveloped the soil matrix. The bottom chip was then replaced with 0.8% agar containing 2',7'-dichlorofluorescein (H₂DCFDA). After moistening the soil and roots, the root-soil interface was imprinted onto the agar for 1 hour. Green fluorescence was observed after the careful removal of the soil. **b**, Representative images. The left panel shows the root-present condition; the right panel shows the root-free control. Scale bar, 100 μm.

[Comment 7] line 77. I do not think that the term “agricultural microbes” is valid. Please rephrase.

Response: We agree that the term “agricultural microbes” is ambiguous. We have now replaced it with “beneficial and pathogenic microbes” to more accurately describe the microbial groups under study. (line 80)

[**Comment 8**] line 146-148. It would be elegant to show this induced expression of e.g. *hemL* in a soil system by qPCR under complex conditions, to support the findings in *in vitro* assays.

Response: We fully agree that demonstrating gene induction in a soil system would strengthen our conclusions. However, due to the difficulty in extracting high-quality RNA from *P. anguilliseptica* in soil, we performed this experiment on agar medium. We isolated *P. anguilliseptica* from wild-type and *rbohD* mutant roots and performed qPCR to assess the expression of chemotaxis-related and *hemL* genes in *P. anguilliseptica*. Our results showed that bacteria colonizing *rbohD* roots showed significantly lower expression of chemotaxis-related genes (*aer2*, *cheA3*), motility-associated gene (*flgE*), and 5-ALA-related gene (*hemL*) than those from wild-type roots (please see Fig. 3g, 4e below). These findings support the notion that rhizosphere ROS facilitate *P. anguilliseptica* colonization *in vivo*.

Fig. 3. *Pseudomonas anguilliseptica* exhibits chemotaxis toward H₂O₂ (100 nM). g, Effect of *P. anguilliseptica* on the expression of chemotaxis and motility-related genes in the rhizosphere of Col-0 and *rbohD*. Gene expression was detected by qRT-PCR and normalized to that of *rpoA*. Data are the mean \pm SD (n = 3). Different letters indicate significant differences among the groups (p < 0.05, one-way ANOVA).

Fig. 4. H₂O₂ (100 nM) promote the synthesis of 5-aminolevulinic acid (5-ALA) in *Pseudomonas anguilliseptica*. e, The relative expression of *hemL* in *P. anguilliseptica* within the rhizosphere of Col-0 and *rbohD*. Gene

expression was detected by qRT-PCR and normalized to that of *rpoA*. Data are the mean \pm SD (n = 3, Student's *t*-test, **p \leq 0.01).

[Comment 9] line 151-154. I might have misunderstood this, but I read the figures as *P. anguillisepta* significantly enhance ROS accumulation after both 1 and 6hpi. Please clarify.

Response: To better highlight the comparative response between *P. anguilliseptica* and *D. solani*, we have focused our analysis on the 1 hpi time point in the revised manuscript. The text has been modified as follows:

line 186-191: “In wild-type plants, *P. anguilliseptica* inoculation induced a modest yet statistically significant ROS accumulation (~20% increase) at 1 hpi, as visualized by 3,3'-diaminobenzidine (DAB) staining (**Fig. 5a-c**). In contrast, *rbohD* mutants showed no detectable ROS induction under the same conditions (**Fig. 5a-c**). Notably, the magnitude of ROS accumulation was substantially weaker than the robust responses typically elicited by pathogens or immune elicitors (>2-fold increase)^{21,28}.”

Reference:

21. Entila, F., Han, X., Mine, A., Schulze-Lefert, P. & Tsuda, K. Commensal lifestyle regulated by a negative feedback loop between *Arabidopsis* ROS and the bacterial T2SS. *Nat. Commun.*, **15**, 456 (2024).
28. Torres, M. A., Jones, J. D. G. & Dangl, J. L. Reactive oxygen species signaling in response to pathogens. *Plant Physiol.*, **141**, 373-378 (2006).

[Comment 10] line 224. The discussion on influence of ROS on metabolic pathways of plants and microbes, and how this could shape a unique microbiome supporting plant health, would benefit from a few sentences on how this would look in a more complex system.

Response: We have added the discussion:

line 269-271: “ROS can directly modulate central metabolic pathways like glycolysis and phenylpropanoid biosynthesis which may selectively enrich beneficial microbes that enhance nutrient acquisition and prime defense responses^{48,49}.”

References:

48. Pinheiro, C. H. D. J. *et al.* Regulation of glycolysis and expression of glucose metabolism-related genes by reactive oxygen species in contracting skeletal muscle cells. *Free Radic. Biol. Med.*, **48**, 953-960 (2010).
49. Yaqoob, U., Jan, N. *et al.* Crosstalk between brassinosteroid signaling, ROS signaling and phenylpropanoid pathway during abiotic stress in plants: Does it

exist? *Plant Stress*, **4**, 100075 (2022).

[**Comment 11**] line 235-236 The sentence does not make sense and needs revision.

Response: In the revised manuscript, we have deleted the perspective section and summarize the key findings and agricultural implications of our study as follows:

line 279-287: “In summary, our study reveals a previously underappreciated signaling role of rhizosphere ROS in orchestrating the selective recruitment of beneficial microbiota. By linking the molecular mechanisms of plant immunity with the dynamics of microbial community assembly, these findings illuminate a chemical dialogue through which plants actively shape their root-associated microbiome. This work not only enhances our mechanistic understanding of plant-microbe interactions but also highlights the potential of harnessing ROS-mediated microbial recruitment as a strategy for microbiome engineering. Ultimately, leveraging these ROS-microbiota interactions could enable the development of sustainable approaches to disease management and crop improvement, offering new opportunities to integrate plant immune regulation with ecological microbiome manipulation in agricultural systems.”

[**Comment 12**] line 291. Please provide the original reference for the biofilm assay, instead of ref 58.

Response: We have now cited the original reference for the biofilm assay. The updated citations are:

62. **Stepanović, S. et al.** A modified microtiter-plate test for quantification of staphylococcal biofilm formation. *J. Microbiol. Methods*, **40**, 175-179 (2000).
63. **Bhowmik, B. et al.** Biofilm associated growth inhibition of XDR escherichia fergusonii strain ACE12 isolated from soil. *Microb. Pathog.*, **201**, 107400 (2025).

[**Comment 13**] Figures: The pink color used throughout the figures can be difficult to read, especially in texts, and I suggest changing the color.

Response: In response, we have revised the figures to replace the pink color with a more contrasting and accessible color scheme, particularly in textual annotations and legends. We now use a clearer red-blue palette to ensure visual clarity and accommodate color-blind readers.

[**Comment 14**] Figure 1. the legend for 1f does not make sense - what does “key taxa between Col-0 and *rbohD*” mean. Please clarify.

Response: We thank you for pointing out the ambiguity in the figure legend. We have revised the legend for Figure 1f as

line 605-606: “The differential taxa between Col-0 and *rbohD* at the genus level based on LDA Effect Size score.”

Dear Dr. Liang,

Thank you for submitting a revised version of your manuscript to The EMBO Journal. We have now received input from two of the original reviewers, who are broadly satisfied with the performed revisions and now request only minor revisions that would mainly require textual adjustments, apart from point 1 by reviewer #1, which could require additional experimentation.

There additionally remain only a few editorial points that need to be addressed before I can extend official acceptance of the manuscript:

1. Please submit a complete author checklist, which you can download from our author guidelines (<https://media.springernature.com/original/springer-cms/rest/v1/content/27825796/data/v1>). Please insert information in the checklist that is also reflected in the manuscript. The completed author checklist will also be part of the Review Process File.
2. Please submit keywords for your manuscript.
3. Please upload the main and EV figures as individual production quality figure files in the .eps, .tif, or .jpg format (one file per figure). Please leave their legends in the manuscript text file, after References.
4. Please rename extended data figures into Figure EV1 - EV11 and assemble their legends after those of main figures under the heading "Expanded View Figure Legends".
5. In the title page, please remove the list of email addresses for the contributing authors.
6. CRedit has replaced the traditional author contributions section because it offers a systematic, machine-readable author contributions format that allows for more effective research assessment. Please remove the Authors Contributions from the manuscript and use the free text boxes beneath each contributing author's name in our online submission system to add specific details on the author's contribution. More information is available in our guide to authors.
7. Please rename "Competing interests" section into "Disclosure and competing interests statement".
8. Please update references according to The EMBO Journal style - where there are more than 10 authors on a paper, the first 10 should be listed, followed by 'et al.' Please remove DOIs from the reference list and add a heading "References" to this section.
9. Please remove suppl. table 2 from the manuscript text, upload it as a separate file and rename it Table EV1. Please update the callouts in the manuscript text accordingly.
10. Rename the uploaded csv file into Dataset EV1 and add a legend with the table's name and a short description to the csv file in a separate tab/worksheet.
11. In the Data Availability section, please add a resolvable link to the PRJNA1215964 dataset.
12. Please remove BioRender disclaimer from the legend for Figure 7 and add to a dedicated section in the Methods section using the following format:

Graphics:

(some of the... OR Figure #... OR synopsis) Graphics were created with BioRender.com.

13. All Materials and Methods need to be described in the main text using our 'Structured Methods' format. According to this format, the Methods section includes a Reagents and Tools Table (listing key reagents, experimental models, software and relevant equipment and including their sources and relevant identifiers) followed by a Methods and Protocols section describing the methods, ideally using a step-by-step protocol format. The aim is to facilitate adoption of the methodologies across labs. Please download and fill our Reagents and Tools Table template (.docx), which you can find in our author guidelines: https://www.embopress.org/pb%2Dassets/embo-site/Reagents_Tools_Table_TEMPLATE.docx

14. Our data editors have flagged the following issues in figure legends that need correcting:

- Please provide the exact p values in the legends of figures 2a,e; 3b,c,f,g; 4d-h; 5b,c,e; 6b,c,d.
- Please indicate the statistical test used for data analysis in the legend of figure 3d.
- Please provide information on the number and nature of replicates in the legend of figure 3d.
- Please describe the nature of replicates in the legends of figures 3b,c,f,g; 4d,e,f.

15. At EMBO Press we ask authors to provide source data for the main manuscript figures. You will receive a separate email with instructions for providing source data with your revised manuscript, including how to upload and organise the files.

16. Papers published in The EMBO Journal are accompanied online by a 'Synopsis' to enhance discoverability of the manuscript. It consists of A) a short (1-2 sentences) summary of the findings and their significance, B) 3-4 bullet points highlighting key results (the highlights can be repurposed for this) and C) a synopsis image that is 550x300-600 pixels large (width x height, jpeg or png format). You can either show a model or key data in the synopsis image. Please note that the image size is rather small and that text needs to be readable at the final size.

17. As part of the EMBO Press transparent editorial process, The EMBO Journal will publish online a Peer Review File to accompany accepted manuscripts. This file will be published in conjunction with your paper and will include the anonymous referee reports, your point-by-point response and all pertinent correspondence relating to the manuscript, including decision letters. Please note that the Author Checklist will be published at the end of the Peer Review File.

Please let us know if you want to remove or not any figures or data from the Peer Review File prior to publication. Please note

that retaining unpublished data in the Peer Review File means that these count as published and that the Peer Review File would need to be referenced in future publications.

With best wishes,

Ieva

Read our guidance for manuscript revisions and related editorial policies: <https://link.springer.com/journal/44318/submission-guidelines#cms-Revised-submissions>

<https://media.springernature.com/original/springer-cms/rest/v1/content/27825798/data/v1>

Please remember: Digital image enhancement is acceptable practice, as long as it accurately represents the original data and conforms to community standards. If a figure has been subjected to significant electronic manipulation, this must be noted in the figure legend or in the 'Methods' section. The editors reserve the right to request original versions of figures and the original images that were used to assemble the figure.

We realize that it is difficult to revise to a specific deadline. In the interest of protecting the conceptual advance provided by the work, we recommend a revision within 3 months (21st Feb 2026). Please discuss the revision progress ahead of this time with the editor if you require more time to complete the revisions.

Referee #1:

The study provides interesting observations on the interaction between *P. anguilliseptica* and *D. solani* in *Arabidopsis*. Although it is unfortunate that genetic manipulation of the *P. anguilliseptica* strain was not feasible, the authors have made reasonable efforts to strengthen their conclusions using the available approaches.

However, several issues remain to be addressed:

1. Recruitment vs. protection

While *P. anguilliseptica* protected Col-0 plants from *D. solani*, this result does not necessarily indicate that *P. anguilliseptica* is recruited to Col-0 more than to *rbohD* plants. The authors should quantify *P. anguilliseptica* abundance under soil conditions (Extended Data Fig. 11) to substantiate this point.

2. Definition of "pathogen-induced ROS"

Although I agree that ROS plays a role in recruiting *P. anguilliseptica*, it remains unclear whether this recruitment is specifically

driven by "pathogen-induced" ROS. The authors demonstrate that MAMP- or DAMP-induced ROS can also recruit *P. anguilliseptica* (Fig. 5). Therefore, the recruitment is not necessarily specific to pathogen-induced ROS. The authors are encouraged to revise the title accordingly.

3. Use of "chemoattractant"

As no bacterial genetic evidence is presented, the use of "chemoattractant" in the title is not fully justified. It would be appropriate to remove or qualify this term.

Additional comments

- Lines 736-737: The figure legend should include detailed inoculation methods and experimental conditions for panels a and b.
- Fig. 2a: The legend should clearly indicate which treatments were inoculated and which were non-inoculated.
- The description of the bacterial gene expression analysis performed in vivo is ambiguous. It is described as "in situ" but appears to involve isolated samples. Please clarify this in both the figure legend and the Methods section.

Referee #2:

This paper is a revision of a manuscript I had previously reviewed. I was already positive about the first submission and the authors have answered all my comments, hence I fully support the publication of the manuscript in the present form.

This work presents novel findings on the role of plant-emitted ROS in mediating changes in bacterial communities associated with plant roots, with profound effects on plant growth and health. ROS thereby act as signals for the recruitment of plant-beneficial microbes, directly enhancing their motility and chemotaxis towards the plant.

I suggest only few minor changes in the abstract and beginning of the main text:

- L1: I would reformulate into "induces the production of respiratory (...) in roots, which diffuse into the rhizosphere.
- L11: inoculation of *P. anguilliseptica* (rather than supplementation with, which is more appropriate for chemical compounds than for living organisms)
- L39: plant-microbe interactions (rather than plant-pathogen)
- L46: beneficial microbiota members or beneficial microbes (but not beneficial microbiota).

Referee #1:

The study provides interesting observations on the interaction between *P. anguilliseptica* and *D. solani* in *Arabidopsis*. Although it is unfortunate that genetic manipulation of the *P. anguilliseptica* strain was not feasible, the authors have made reasonable efforts to strengthen their conclusions using the available approaches.

However, several issues remain to be addressed:

[Comment 1] 1. Recruitment vs. protection

While *P. anguilliseptica* protected Col-0 plants from *D. solani*, this result does not necessarily indicate that *P. anguilliseptica* is recruited to Col-0 more than to *rbohD* plants. The authors should quantify *P. anguilliseptica* abundance under soil conditions (Extended Data Fig. 11) to substantiate this point.

Response: We appreciate the reviewer's insightful comment. To address this point, we quantified the abundance of GFP-labeled *P. anguilliseptica* in the rhizosphere of Col-0 and *rbohD* plants under soil-grown conditions. Consistently, the *rbohD* mutant exhibited a lower abundance of *P. anguilliseptica* in its rhizosphere. These results are presented in Fig EV5 of the revised manuscript (also see figure below), with the following description added:

line 226-229: "Soil-based co-inoculation experiments further showed that *P. anguilliseptica* alleviated *D. solani*-induced disease symptoms in wild-type plants but not in *rbohD* mutants (Fig EV5A and B). Consistently, lower abundance of *P. anguilliseptica* was observed in the rhizosphere of *rbohD* mutants (Fig EV5C)."

Fig EV5. *Pseudomonas anguilliseptica* promotes plant resistance against *Dickeya solani* infection.

C Rhizosphere colonization of *P. anguilliseptica*. Three-week-old Col-0 and *rbohD* plants were inoculated with GFP-labeled *P. anguilliseptica* alone or

co-inoculated with *D. solani* at equal cell densities ($OD_{600} = 0.5$, 15 mL per plant). One-week post-inoculation, rhizosphere soil was collected, and bacterial populations were determined by plating. Data represent mean \pm SD ($n = 8$ biological replicates). P values are determined (Student's *t*-test).

We have additionally provided the experimental details in the Methods section. (line 365-368)

[Comment 2] 2. Definition of "pathogen-induced ROS"

Although I agree that ROS plays a role in recruiting *P. anguilliseptica*, it remains unclear whether this recruitment is specifically driven by "pathogen-induced" ROS. The authors demonstrate that MAMP- or DAMP-induced ROS can also recruit *P. anguilliseptica* (Fig. 5). Therefore, the recruitment is not necessarily specific to pathogen-induced ROS. The authors are encouraged to revise the title accordingly.

Response: We have deleted "pathogen-induced" and revised the title to "Reactive oxygen species in the rhizosphere orchestrate the recruitment of beneficial bacteria".

[Comment 3] 3. Use of "chemoattractant"

As no bacterial genetic evidence is presented, the use of "chemoattractant" in the title is not fully justified. It would be appropriate to remove or qualify this term.

Response: We have removed the term "chemoattractant" and revised the title to "Reactive oxygen species in the rhizosphere orchestrate the recruitment of beneficial bacteria".

[Comment 4] Lines 736-737: The figure legend should include detailed inoculation methods and experimental conditions for panels a and b.

Response: We have updated the figure legend for panels A and B to include detailed inoculation methods and all relevant experimental conditions. (Appendix line36-40)

Appendix Figure S4. *rbohD* mutants exhibit increased susceptibility to *Dickeya solani* compared to the wild-type.

A-E Disease symptoms caused by *D. solani* in mature plants. Three-week-old plants in soil was inoculated with *D. solani* ($OD_{600} = 0.5$, 15 mL). One week post-inoculation, phenotypes were recorded (A). Ten-day-old seedlings were

inoculated with *D. solani* ($OD_{600} = 0.001$). Representative images, seedling survival rate, and fresh weight are shown in (B), (C), and (D), respectively. The bacterial populations were quantified by flow cytometry at 3 dpi (E). Scale bar, 1 cm. Box plots display the median, upper, and lower quartile, with whiskers extending up to 1.5 times the interquartile range ($n = 13$ biological replicates). P values are determined (Student's t-test).

[**Comment 5**] Fig. 2a: The legend should clearly indicate which treatments were inoculated and which were non-inoculated.

Response: We have clarified the inoculation status in Fig. 2A by labeling the x-axis with the names of the bacterial strains together with plus/minus signs to indicate inoculated and non-inoculated treatments.

Fig 2. RBOHD-derived ROS contribute to the enrichment of beneficial bacteria in the rhizosphere.

A *Pseudomonas anguilliseptica*-induced enhancement of seedling fresh weight in Col-0, but not in *rbohD*. Box plots display the median, upper, and lower quartiles, with whiskers extending up to 1.5 times the interquartile range ($n = 16$ biological replicates). P values are determined (ANOVA and Tukey test).

[**Comment 6**] The description of the bacterial gene expression analysis performed in vivo is ambiguous. It is described as "*in situ*" but appears to involve isolated samples. Please clarify this in both the figure legend and the Methods section.

Response: In the revised manuscript, we have removed the term "*in situ*" from the description and clarified the corresponding experimental details in the Methods section. (line 375-376)

Referee #2:

This paper is a revision of a manuscript I had previously reviewed. I was already positive about the first submission and the authors have answered all my comments, hence I fully support the publication of the manuscript in the present form.

This work presents novel findings on the role of plant-emitted ROS in mediating changes in bacterial communities associated with plant roots, with profound effects on plant growth and health. ROS thereby act as signals for the recruitment of plant-beneficial microbes, directly enhancing their motility and chemotaxis towards the plant.

I suggest only few minor changes in the abstract and beginning of the main text:

[Comment 1] L1: I would reformulate into "induces the production of respiratory (...) in roots, which diffuse into the rhizosphere.

Response: As suggested by Reviewer #1, we have removed "pathogen-induced" from the title. To maintain consistency with this change, we have revised the opening sentence of the abstract to the following:

"Although respiratory burst oxidase homolog D (RBOHD)-dependent reactive oxygen species (ROS) in *Arabidopsis* are well known to suppress pathogen colonization, their influence on beneficial microbes remains unclear."

We believe this version presents the scientific question in a more focused and concise manner.

[Comment 2] L11: inoculation of *P. anguilliseptica* (rather than supplementation with, which is more appropriate for chemical compounds than for living organisms)

Response: We have revised the phrasing to "inoculation of *P. anguilliseptica*". (line 11)

[Comment 3] L39: plant-microbe interactions (rather than plant-pathogen)

Response: We have revised the phrasing to "plant-microbe interactions". (line 43)

[Comment 4] L46: beneficial microbiota members or beneficial microbes (but not beneficial microbiota).

Response: We have replaced the previous text with the phrase "beneficial microbiota

members". (line 49)

Dear Dr. Liang,

Thank you for submitting the final revised version and addressing the remaining points. I am now pleased to inform you that your manuscript has been accepted for publication.

Before we forward your manuscript to our publishers, we would like to propose some edits in the manuscript's abstract and synopsis (please see the attached file). I have also prepared a short blurb that will accompany the title of your manuscript in our online system. Please take a look and let me know if any corrections are needed.

Please note that it is The EMBO Journal policy for the transcript of the editorial process (containing referee reports and your response letters) to be published as an online supplement to each paper. If you should prefer removal of any referee-only figures included in the point-by-point response(s), e.g. because they may still be used for future publication or because they have been reproduced from published work by others, please do let us know immediately via response email.

More information is available here: <https://link.springer.com/partners/embo-press/editorial-policies#Peer%20review>

You may qualify for financial assistance for your publication charges - either via a Springer Nature fully open access agreement or an EMBO initiative. Check your eligibility: <https://link.springer.com/journal/44318/how-to-publish-with-us>

If you have any questions, please do not hesitate to contact the Editorial Office or me directly. Thank you for this contribution to The EMBO Journal and congratulations on a nice study!

With best wishes,

leva

leva Gailite, PhD
Senior Scientific Editor
The EMBO Journal
Meyerhofstrasse 1
D-69117 Heidelberg
Tel: +4962218891309
i.gailite@embojournal.org